# Presenting a sham treatment as personalised increases the placebo effect in a randomised controlled trial

**Dasha A Sandra[1]\*, Jay A Olson[2†], Ellen J Langer[2], Mathieu Roy[3]**

[1]Integrated Program in Neuroscience, McGill University, Montreal, Canada; [2]Department of Psychology, Harvard University, Cambridge, United States; [3]Department of Psychology, McGill University, Montreal, Canada

## Abstract

**Background:** Tailoring interventions to patient subgroups can improve intervention outcomes for various conditions. However, it is unclear how much of this improvement is due to the pharmacological personalisation versus the non-specific effects of the contextual factors involved in the tailoring process, such as the therapeutic interaction. Here, we tested whether presenting a (placebo) analgesia machine as personalised would improve its effectiveness.

**Methods:** We recruited 102 adults in two samples ($N_1$=17, $N_2$=85) to receive painful heat stimulations on their forearm. During half of the stimulations, a machine purportedly delivered an electric current to reduce their pain. The participants were either told that the machine was personalised to their genetics and physiology, or that it was effective in reducing pain generally.

**Results:** Participants told that the machine was personalised reported more relief in pain intensity than the control group in both the feasibility study (standardised $\beta$=−0.50 [−1.08, 0.08]) and the preregistered double-blind confirmatory study ($\beta$=−0.20 [−0.36, −0.04]). We found similar effects on pain unpleasantness, and several personality traits moderated the results.

**Conclusions:** We present some of the first evidence that framing a sham treatment as personalised increases its effectiveness. Our findings could potentially improve the methodology of precision medicine research and inform practice.

**Funding:** This study was funded by the Social Science and Humanities Research Council (93188) and Genome Québec (95747).

**\*For correspondence:** dasha.sandra@mail.mcgill.ca

**Present address:** †University of Toronto Mississauga, Mississauga, Canada

**Competing interest:** The authors declare that no competing interests exist.

## Editor's evaluation

Sandra et al. assessed the effects of a personalized intervention on the placebo effect in a randomized controlled trial. The study showcases important results highlighting that psychological aspects of 'personalised' or 'precision' medicine substantially shape the treatment effects over and above the benefit of biologically/clinically/pharmacologically tailored interventions. It has to be noted that the effect sizes identified are relatively small and the outcomes are subjective, which has implications for the generalizability of the results.

## Introduction

Precision medicine may revolutionise healthcare by tailoring interventions to patients' specific genetic, biological, and behavioural markers. Targeted therapies can lead to better health outcomes, such as increased life expectancy and remission rates, notably in cancer (*Cutler, 2020*). Researchers are now attempting to extend precision medicine approaches to other conditions such as chronic pain (*Reimer*

**eLife digest** Precision treatments are therapies that are tailored to a patient's individual biology with the aim of making them more effective. Some cancer drugs, for example, work better for people with specific genes, leading to improved outcomes when compared to their 'generic' versions. However, it is unclear how much of this increased effectiveness is due to tailoring the drug's chemical components versus the contextual factors involved in the personalisation process.

Contextual factors like patient beliefs can boost a treatment's outcomes via the 'placebo effect' – making the intervention work better simply because the patient believes it to. Personalised treatments typically combine more of these factors by being more expensive, elaborate, and invasive – potentially boosting the placebo effect.

Sandra et al. tested whether simply describing a placebo machine – which has no therapeutic value – as personalised would increase its effectiveness at reducing pain for healthy volunteers. Study participants completed several sham physiological and genetic tests. Those in the experimental group were told that their test results helped tailor the machine to increase its effectiveness at reducing pain whereas those in the control group were told that the tests screened for study eligibility.

All volunteers were then exposed to a series of painful stimuli and used the machine to reduce the pain for half of the exposures. Participants that believed the machine was personalised reported greater pain relief. Those with a stronger desire to be seen as different from others – based on the results of a personality questionnaire – experienced the largest benefits, but only when told that the machine was personalised.

This is the first study to show that simply believing a sham treatment is personalised can increase its effectiveness in healthy volunteers. If these results are also seen in clinical settings, it would suggest that at least some of the benefit of personalised medicine could be due to the contextual factors surrounding the tailoring process. Future work could inform doctors of how to harness the placebo effect to benefit patients undergoing precision treatments.

*et al., 2021*). Further, advancements in artificial intelligence may soon broaden the use of personalisation for drug dosing (*Rybak et al., 2020*) and treatment selection (*Ahmed et al., 2020*). However, the greater effectiveness of tailored interventions may be due to more than just their pharmacological ingredients: *contextual factors*, such as the treatment setting and patient beliefs, may also directly contribute to better outcomes. The influence of contextual factors in precision medicine remains relatively unexplored, despite experts highlighting their potential influence on intervention outcomes (*Haga et al., 2009*). Thus, isolating the role of the contextual factors involved in the personalisation process could help control for them in precision medicine research and possibly optimise them in clinical practice.

Research in placebo science has shown that the contextual factors surrounding the intervention, such as verbal suggestions, patient expectations, social cues, and observational learning increase its effectiveness and reduce the associated side effects (*Bernstein et al., 2020*; *Colloca and Barsky, 2020*; *Olson et al., 2021a*). Some of these contextual factors can both modulate the effectiveness of inactive treatments in lab settings as well as increase the placebo effect of the real treatments in clinical settings (*Blasini et al., 2018*). Additionally, these effects are present for both subjective symptoms (e.g. pain, depression) and physiological ones (e.g. immune response, motor function) (*Benedetti et al., 2005*). Precision medicine may already benefit from greater patient expectations given the public's high hopes for the field (*Collins and Varmus, 2015*), increased trust, and possible preference to be seen as different from others (i.e., high need for uniqueness), which may all increase placebo effects.

The public generally believes that personalised interventions are fully unique, so much so that the field rebranded from 'personalised' to 'precision' medicine in an effort to dispel this exaggerated view (*Juengst et al., 2016*). Despite the field's more modest focus on targeting patient subgroups, tailored interventions use information about individual genetics and biology—elements most people believe to define their individual essence (*Gelman, 2003*). If an intervention was tailored to something so unique, the treatment would indeed be more likely to work, in turn possibly raising patients' expectations. This appeal may be particularly strong in the broader context of rising individualism (*Santos*

*et al., 2017*) and may speak to patients' desire to be seen as distinct individuals. Although no studies to our knowledge have directly explored the influence of genetic information on perceived treatment effectiveness, receiving sham genetic feedback itself can affect behaviour and physiology, suggesting the potential for placebo effects in precision therapies. For example, simply learning about one's increased genetic risk for obesity may lead to lower self-efficacy, reduced perceived control over related behaviours (*Beauchamp et al., 2011*; *Dar-Nimrod et al., 2014*), and worse cardiorespiratory capacity (*Turnwald et al., 2019*); learning one has a protective genetic makeup may cause the opposite results (*Turnwald et al., 2019*), regardless of the actual genes involved. Providing genetic and physiological feedback and then using it to tailor a treatment may similarly influence outcomes for precision therapies.

Beyond the appeal to individuality, the personalisation process may implicitly suggest stronger perceived treatment effectiveness by harnessing factors well-known to increase the placebo effect (*Olson et al., 2021a*). Pharmacological tailoring is an intricate process and often requires biomarker tests that are sometimes invasive (*Corcoran, 2020*), take longer to process (*Rieder et al., 2005*), or use advanced technology (*Rybak et al., 2020*). Indeed, studies in placebo science show that treatments that are more elaborate, invasive (*de Craen et al., 2000*; *Hróbjartsson and Gøtzsche, 2010*; *Meissner et al., 2013*), or use complex technology may cause larger improvements (*Kaptchuk et al., 2000*; *Kaptchuk et al., 2008*). Given the complexity of the procedure, tailoring treatments requires more physician attention and the involvement of practitioners specifically trained in therapeutic communication, such as genetic counsellors (*Austin et al., 2014*; *Kohut et al., 2019*). Practitioners may also inadvertently suggest greater effectiveness of the treatment by explaining it in more detail. Similarly, placebo studies show that providing enhanced information about a treatment can increase the effects of already potent drugs like opioids (*Amanzio et al., 2001*; *Benedetti et al., 2003*), and positive communication strategies may reduce the side effects of sham pills (*Barnes et al., 2019*; *Colloca and Finniss, 2012*). More broadly, a warm and empathetic encounter can improve outcomes for active and inactive treatments in clinical settings (*Blasini et al., 2018*).

Despite the similarities between the ideal contextual factors for strong placebo effects and the typical contextual factors involved in precision medicine, the influence of the personalisation process on perceived treatment effectiveness is largely unknown. Thus, we tested whether believing that a treatment is tailored to one's physiology and genetics may improve its perceived efficacy. We predicted that participants using the machine presented as personalised would report greater placebo effects than those in a control group.

To isolate the role of the placebo effects of personalisation while avoiding the ethical issues involved in deceiving severely ill patients—the typical participants in precision clinical trials—we tested healthy adults. We developed an elaborate procedure to plausibly simulate treatment personalisation and then tested it in a feasibility study ($N_1$=17) before confirming the findings in a pre-registered double-blind experiment ($N_2$=85). The procedure was based on studies of complex placebo interventions (*Olson et al., 2021a*; *Olson and Raz, 2021c*) and simulated both the nature of the tests (i.e., genetic, physiological) and the medical context (i.e., room setting, location) of treatment tailoring. We also measured several personality traits that could potentially interact with the placebo effects of personalisation. Recent studies suggest that traits such as interoceptive awareness (attention to one's physical sensations) and openness to experience predict the magnitude of placebo response (*Vachon-Presseau et al., 2018*); we additionally expected that other traits such as need for uniqueness (the desire to be seen as different from others) may moderate the specific placebo effects of personalisation.

## Materials and methods
### Feasibility study
#### Participants
We recruited 19 participants aged 18–35 from the McGill University community. One person was excluded due to technical errors during testing and another one for guessing the placebo component. The final sample included 17 participants (14 women) who were undergraduate psychology students (n=9) and 21.1 years old on average (SD = 2.9). Most participants were White (n=6) or Asian (n=6). The study was approved by the McGill University Research Ethics Board II (#45–0619).

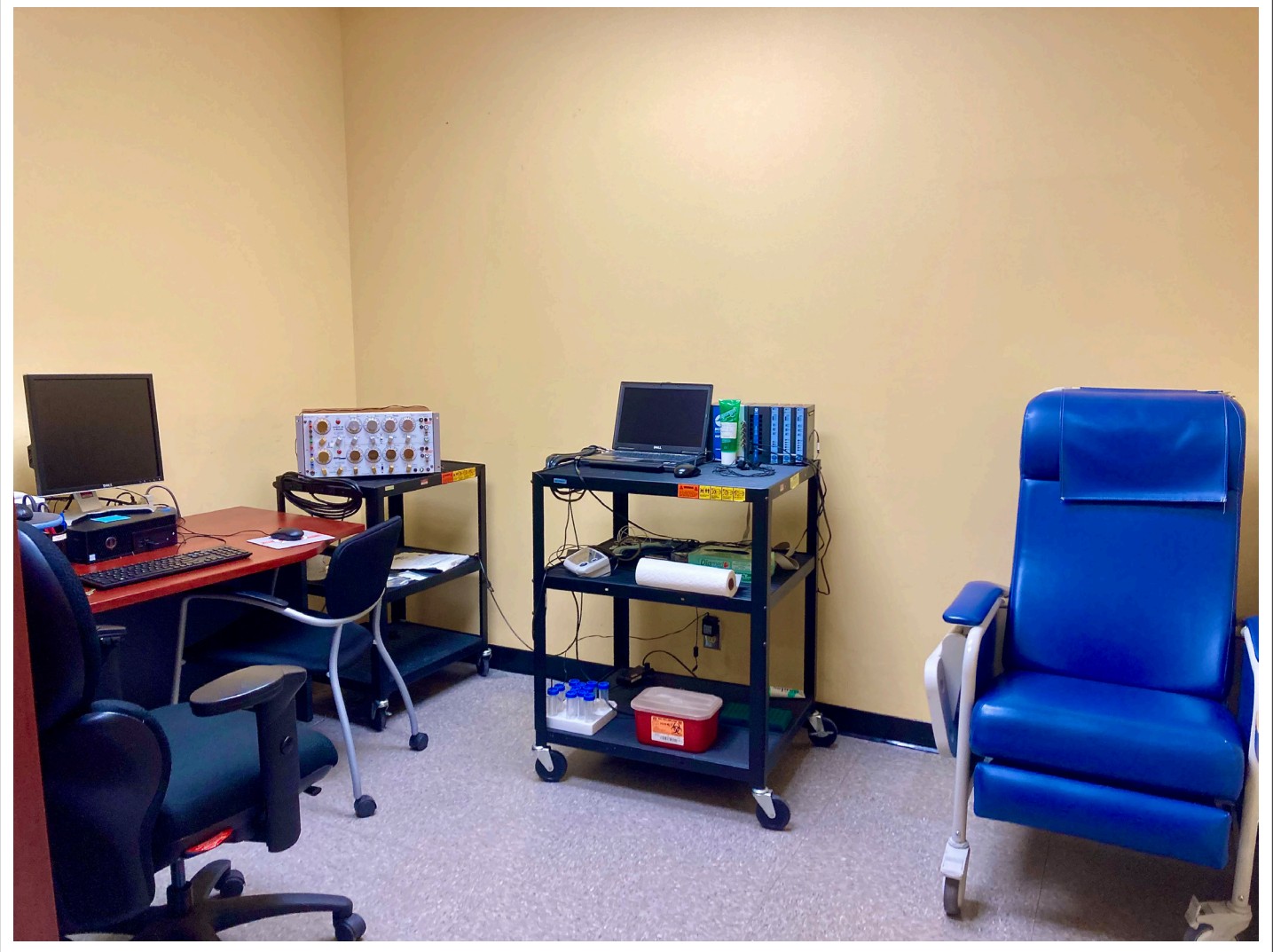

**Figure 1.** Participants completed sham medical tests and then rated pain stimulations in a room with various medical equipment.

## Procedure

Before arriving at the lab (*Figure 1*), participants consented to participate in the study and completed several personality questionnaires: the Need for Uniqueness Scale (*Snyder and Fromkin, 1977*), Multidimensional Assessment of Interoceptive Awareness (*Mehling et al., 2012*), Big Five Inventory (*John and Srivastava, 1999*), Fear of Pain Questionnaire-III (*McNeil and Rainwater, 1998*), and the Pain Catastrophizing Scale (*Sullivan et al., 1995*); see SI Appendix for descriptions. Once at the lab, participants met two female experimenters at a medical building of a large Canadian university. The experimenters introduced themselves as neuroscience researchers and explained the study procedure. Participants learned about the study and were introduced to the placebo machine (*Figure 2*), which was presented as an analgesic device used in hospitals.

## Pain calibration

The experimenter then calibrated participants' individual levels of pain for the pain task (*Tabry et al., 2020*); the calibration was performed once. The experimenter marked four 3 cm long locations on the participants' inner forearm and then applied heat to each of these in a random order using the Medoc Pathway heat stimulator (3×4 cm, TSA-II Neurosensory Analyzer, Medoc Advanced Medical Systems Ltd., Ramat Yishai, Israel). Participants completed 28 heat stimulations: 7 temperatures per spot, ranging from 40 °C to 49 °C, generating the participant's pain sensitivity curve. Each heat

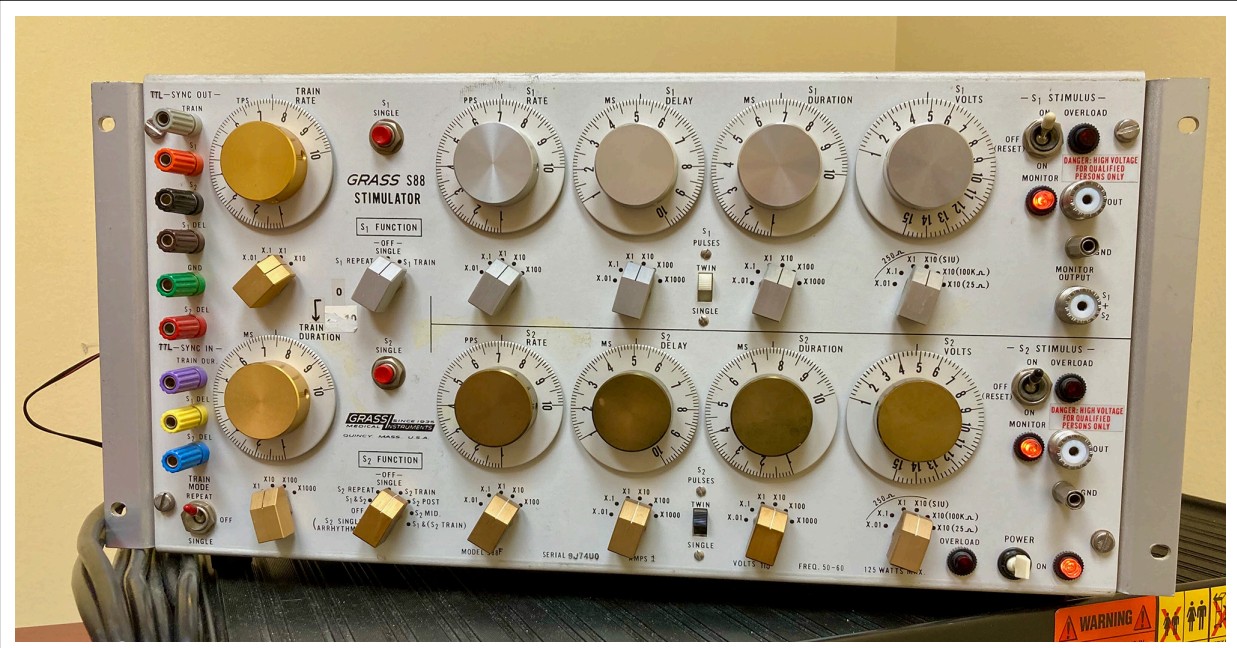

**Figure 2.** On half of the stimulations, participants used a complex placebo machine with dials, vibration, and flashing lights to help reduce pain. This machine was presented as either personalised to their test results or as generally effective. The machine's design (over a dozen of switches and dials) allowed us to simulate complex personalisation to the participants' profile.

stimulation lasted 9 s (2.5 s ramp-up, 4 s maximum temperature, and 2.5 s ramp-down at the rate of 2.3 °C/s). Participants rated the stimulation as perceived heat or pain: for heat stimulations they rated the warmth on a visual analogue scale (0–100) to determine their pain threshold; for pain they rated the intensity (strength) and unpleasantness (discomfort) on separate scales (0–100) to determine the perception of pain levels. The task took approximately 20 min and was coded in E-Prime (Psychology Software Tools, Inc, Sharpsburg, PA). After its completion, participants were randomised to receive either a 'personalised' placebo machine or not.

### Sham medical tests
All volunteers completed additional sham genetic and electrodermal skin response tests. For the genetic test, participants provided a saliva sample using a commercially available DNA kit. To feign the electrodermal skin response test, the experimenter attached two electrodes to participants' fingers and then pretended to record their galvanic skin response for one minute.

### Personalised group
During the procedure, participants learned that the experimenter would adjust the machine to their test results in an effort to increase its effectiveness. Once the tests were complete, the experimenter provided sham genetic and physiological feedback to the participant, reiterated that these were useful for the machine personalisation, and explained the machine functioning in detail. The experimenter then adjusted several dials and switches on the machine to match the participants' results in front of them. Finally, the participants tested the machine briefly to increase their comfort with it (as well as its believability). For this, the experimenter attached two electrodes to the participants' forearm and connected the machine to them for approximately one minute.

### Control group
Those in the control group completed the same procedure ostensibly for eligibility instead of person-alisation. The experimenter received the participant's genetic feedback and informed the participants that they were eligible for the study. To match the duration of interaction and explanations provided in the personalised group, the experimenter instead described the different kinds of analgesics used

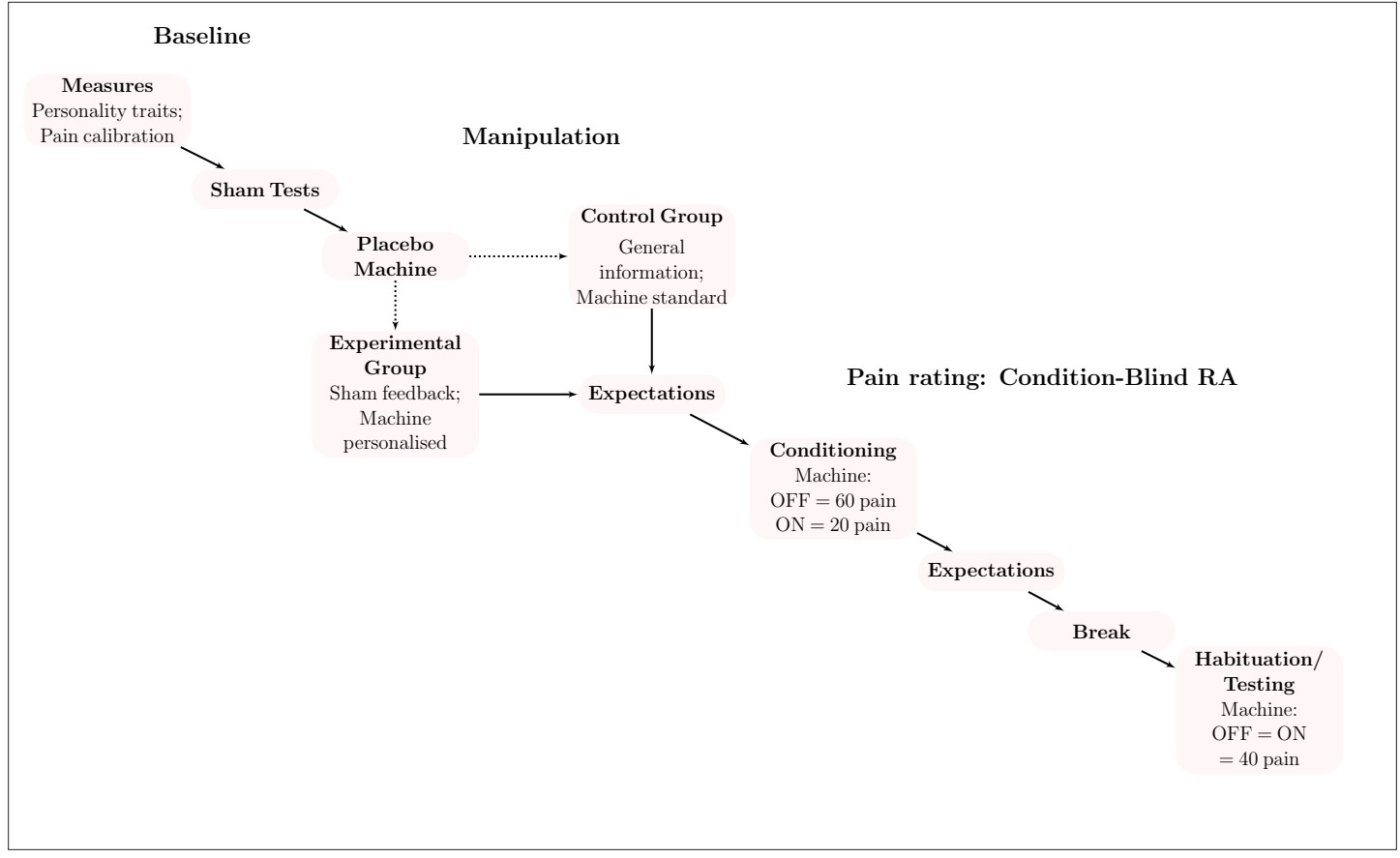

**Figure 3.** Procedure for the confirmatory study. We first asked participants to complete personality questionnaires and calibrated heat stimulations to their individual pain perception. Participants then completed sham medical tests (i.e., genetics, skin conductance) before being randomised to receive the placebo machine described as personalised to their sham test results or not (control). A research assistant blind to the experimental condition then led participants through a pain rating task that was similar to the calibration. On half of the heat stimulations, participants used the machine (turned on) to counteract the heat pain (on the other half, the machine was turned off). In the conditioning phase, we simulated machine effectiveness by covertly reducing the intensity of pain stimulations when the machine was turned on. For the testing phase, we kept the temperature stable and quantified the placebo effect as the difference between the trials with the machine off and on.

in hospitals. The experimenter provided approximately 300 words of information to each group (280 in experimental and 298 in control). Finally, the experimenter introduced the machine with the same description and demonstration.

## Placebo machine

We used a defunct electrical stimulator with various dials, switches, and buttons. The machine had several lights that flashed when turned on and a small vibrating device behind the machine to mimic buzzing. We used a real electric current to increase machine credibility: we hid a small Transcutaneous Electrical Nerve Stimulation (TENS) device behind the placebo machine, which was connected to the electrodes placed on the participant's arm. The device was set at a non-therapeutic intensity that was barely perceptible and was administered for a few minutes at a time, as opposed to at therapeutic levels (i.e., high intensity and for at least 20 min).

## Pain rating task

To reduce demand characteristics, a research assistant blind to the condition replaced the experimenter to run the participant through a validated pain task (*Wager et al., 2004*). The assistant then led the participants through 18 stimulation trials in 3 phases: conditioning (8), habituation (2), and testing (8). The stimulations followed the same procedure as the pain calibration task: participants received heat stimulations lasting 9 s each and rated these stimulations on pain intensity and unpleasantness

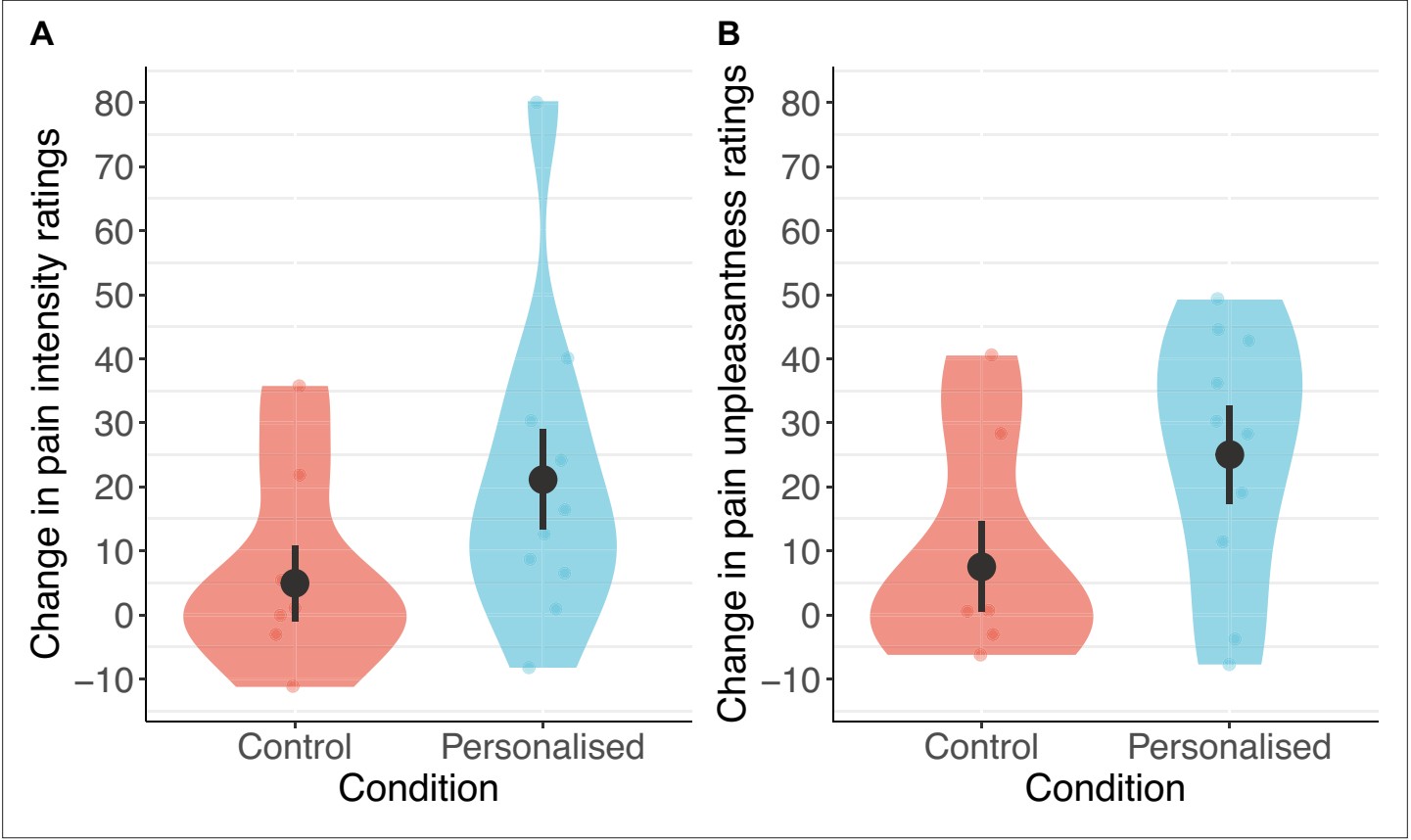

**Figure 4.** Participants in the personalised group reported nearly twice the reduction in pain intensity (**A**) and unpleasantness (**B**; N=17). The placebo effect was calculated as ratings with the machine off – machine on. Black dots show means, coloured dots show individual raw scores, violin widths show frequency, and error bars show 95% confidence intervals.

using the same visual analogue scale coded with similar software (PsychoPy, version 3.1). We used temperatures corresponding to the participants' respective pain levels obtained during the calibration task in order to standardise pain perception across participants. The conditioning phase of the task was used to demonstrate the machine's effectiveness. Participants received 4 pain stimulations at 80/100 level pain when the machine was turned off and 4 stimulations at 20/100 level pain when the machine was turned on, in a counterbalanced order. To minimise habituation and sensitisation noise from the repeated pain stimulations, we applied heat randomly on areas 1 and 3 (out of the 4 spots previously marked) on the participant's arm, and reserved spots 2 and 4 for testing. A 5-min break followed, during which participants completed a filler creativity task (*Olson et al., 2021b*).

After the break, participants completed 2 habituation trials with level 50 pain on areas 2 and 4 of the participant's arm, followed by 8 testing trials on the same spots, with the same on–off order as conditioning and level 50 of heat pain.

### Probing for suspicion
At the end of the study, the experimenter interviewed participants about their experience, probed them for suspicion about the true purpose of the study (*Nichols and Edlund, 2015*), and provided a partial debriefing. All participants were fully debriefed after the end of data collection.

### Confirmatory study
#### Participants
The sample size, exclusion criteria, and analyses were pre-registered online (https://osf.io/dcs98). We recruited 106 healthy participants aged 18–35 from the McGill University community; these were students and recent graduates from various disciplines. Of all participants, 1 did not complete the

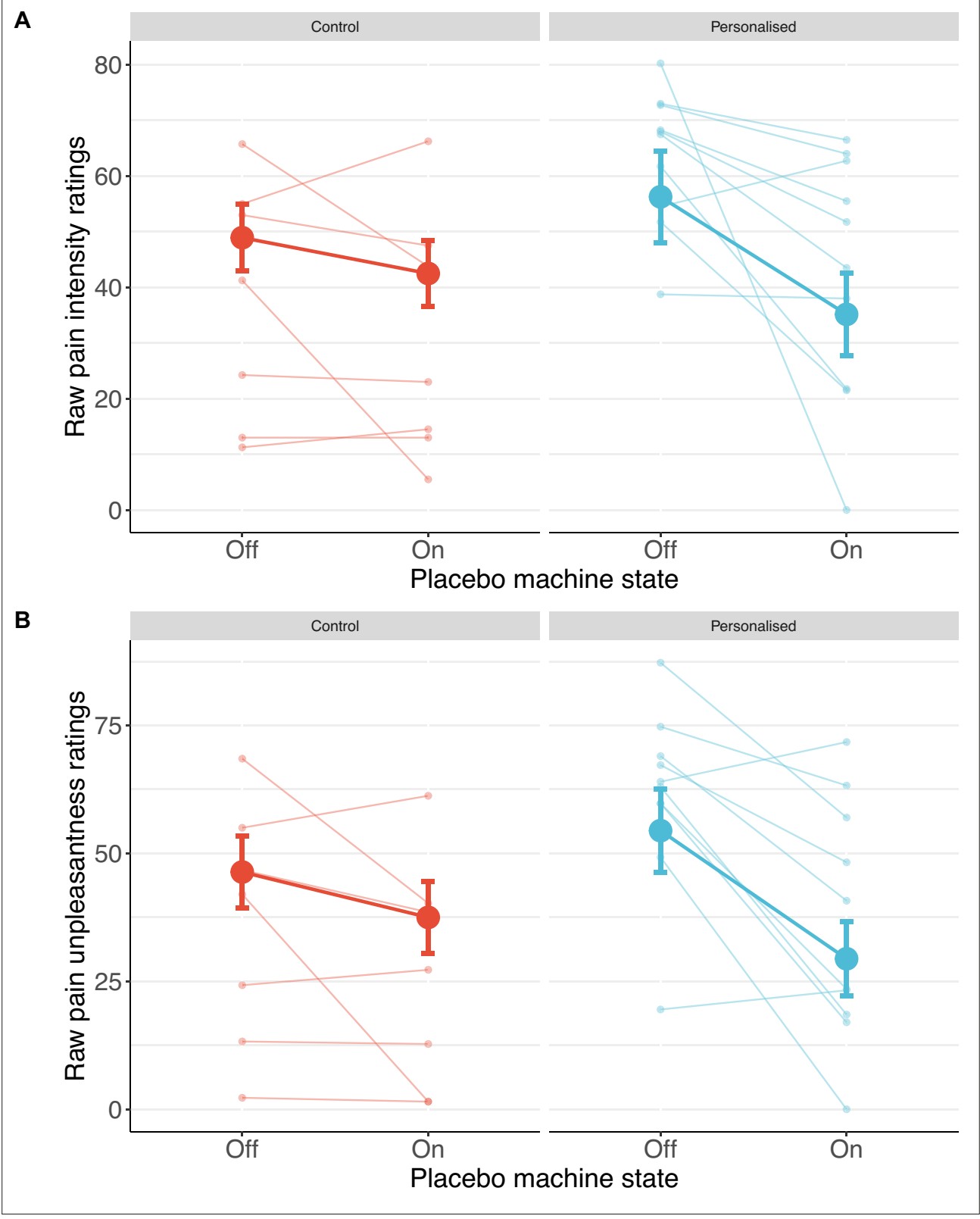

**Figure 5.** Individual pain score changes with the placebo machine turned on or off for pain intensity (**A**) and unpleasantness (**B**). Large coloured dots show means, small coloured dots show individual scores, and error bars show 95% confidence intervals.

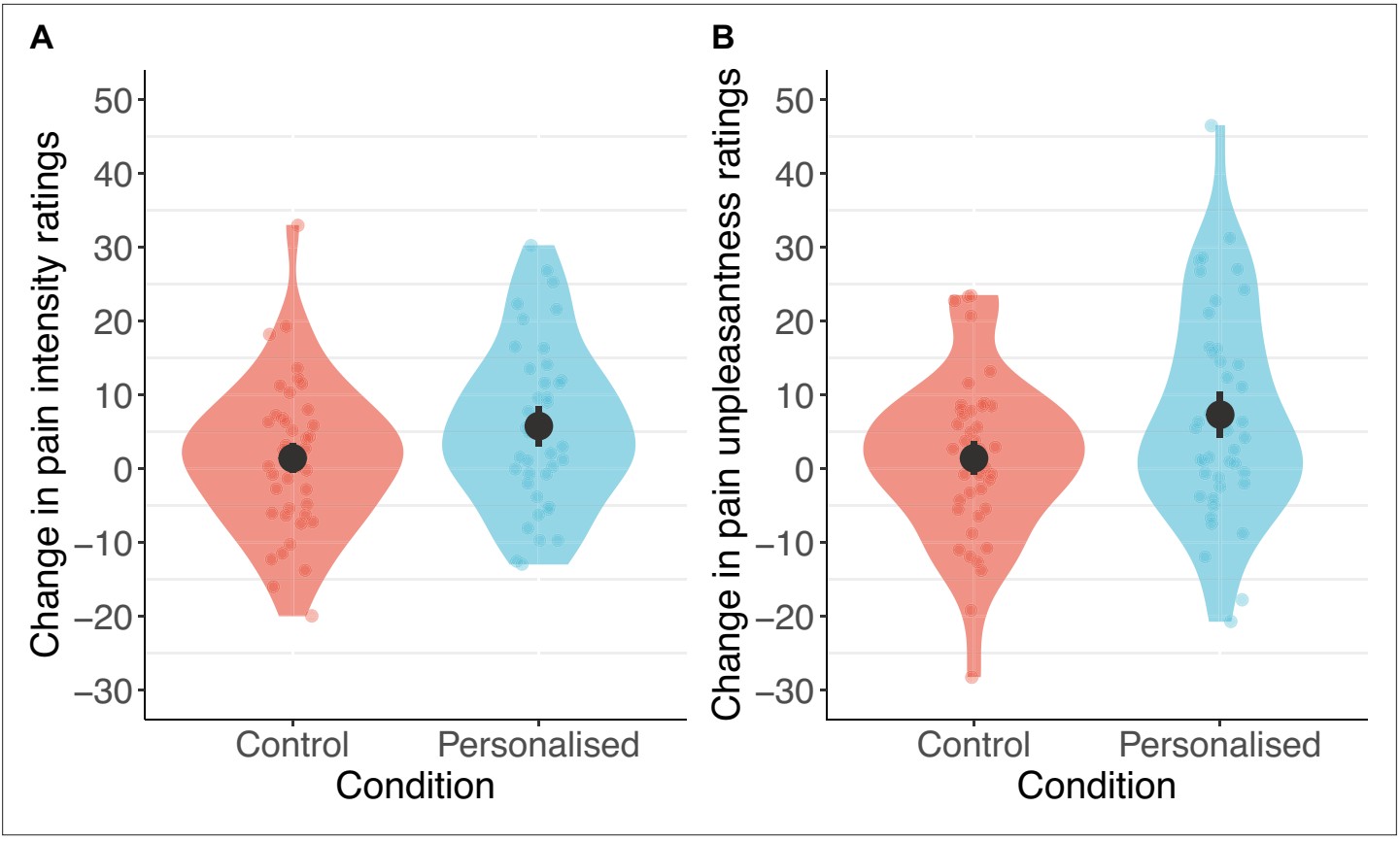

**Figure 6.** Participants in the personalised group reported higher placebo effects than those in the control group for pain intensity (**A**) and unpleasantness (**B**; N=85). The panels show changes calculated as ratings with the machine off – machine on. Black dots show means, coloured dots show individual raw scores, violin widths show frequency, and error bars show 95% confidence intervals.

questionnaires which included the consent form, 6 did not fit eligibility criteria after consenting to participate, 1 experienced technical errors during the experiment, 1 refused to use the machine, and 12 mentioned or asked about the placebo effect (6 in each group). We were stringent with the exclusion criteria to avoid positively biasing our effect: we only excluded participants who explicitly mentioned the placebo effect with additional explanations. For instance, one participant expressed general suspicion about stimulation timings and asked about placebo effects in the beginning of the session and was therefore excluded. The final sample included 85 participants (71 women) with a mean age of 21.4 (SD = 2.2). Most participants were White (n=42) or Asian (n=34). We excluded one additional participant from the analyses of expectations due to missing data. The study was approved by the McGill University Research Ethics Board II (#45–0619).

### Procedure

The procedure and measures were identical to those reported for the feasibility study, with the changes listed below (*Figure 3*).

### Pain task

In this study, we used a pain level of 60 out of 100 for the conditioning-machine-off block (instead of 80 in the feasibility study) and level 40 for the testing blocks (instead of 50). We reduced the gap between off–on temperatures to increase the believability of the machine's effect in the confirmatory study. On average, participants reported a pain threshold of 45.9 °C (SD = 1.7), as well as 46.9 °C (1.3) for pain level 20, 47.8 °C (1.0) for pain level 40, and 48.5 °C (1.2) for pain level 60.

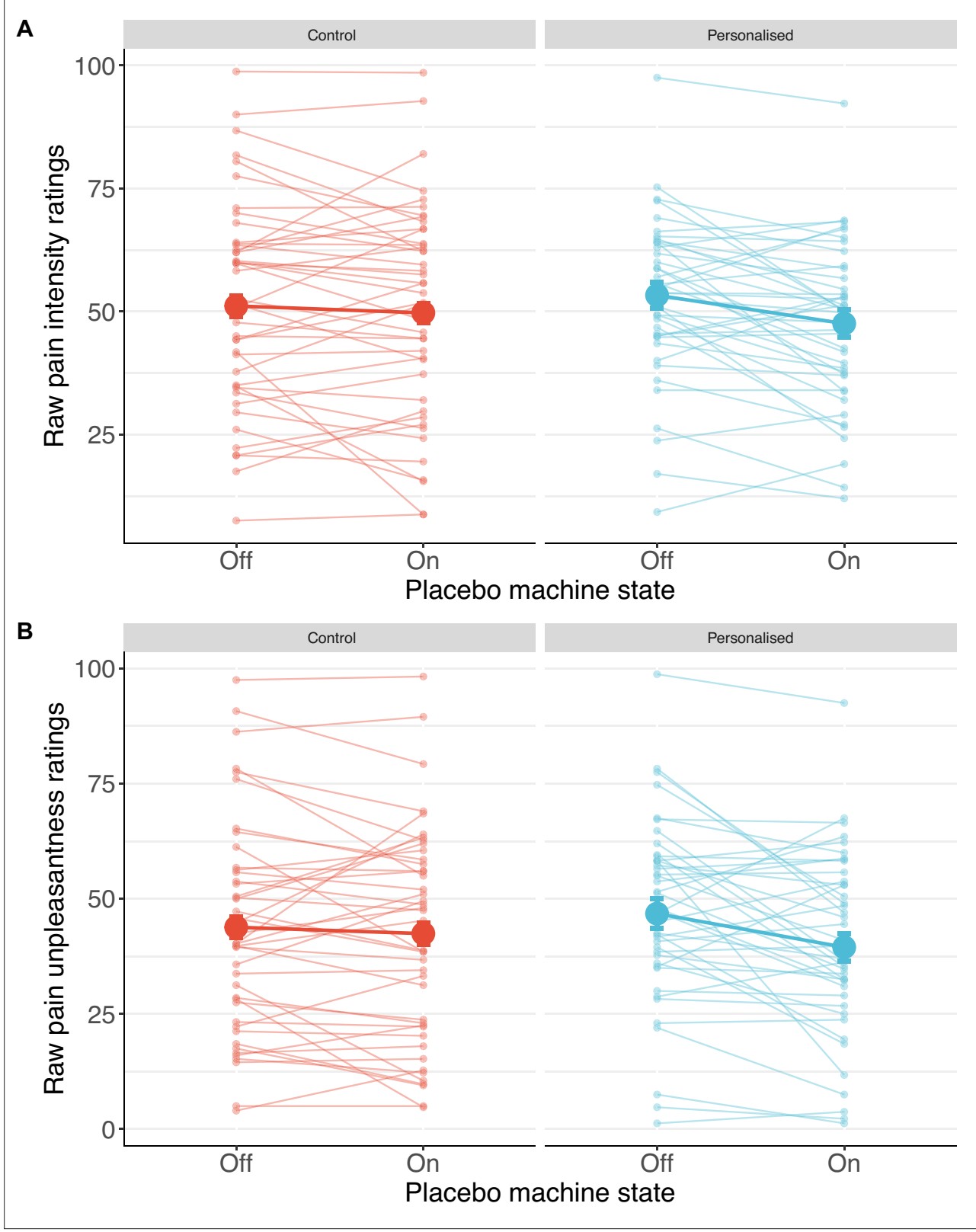

**Figure 7.** Individual pain score changes with the placebo machine turned on or off for pain intensity (**A**) and unpleasantness (**B**). Large coloured dots show means, small coloured dots show individual scores, and error bars show 95% confidence intervals.

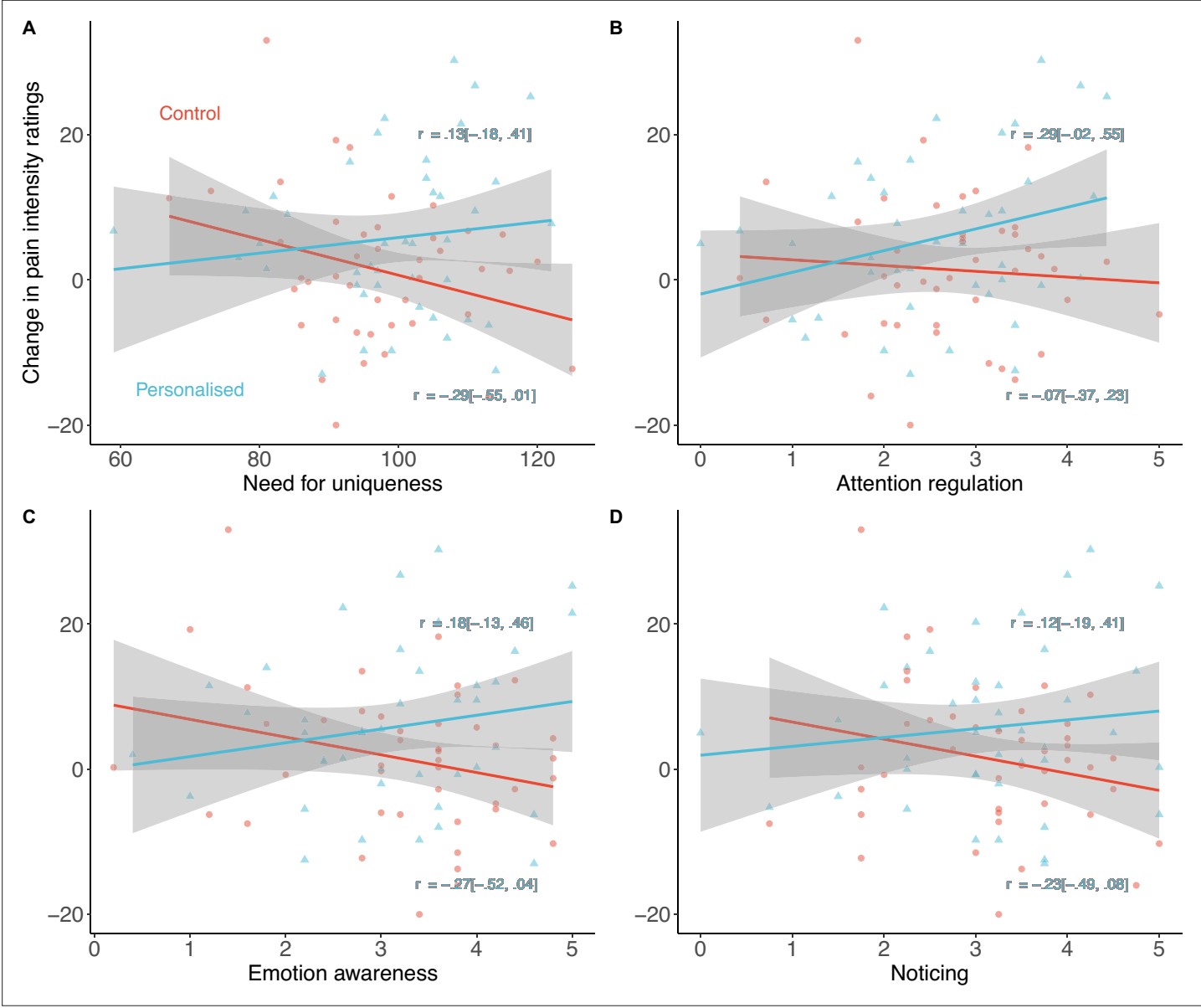

**Figure 8.** Exploratory predictors of placebo effects on pain intensity (N=85). Participants high in Need for uniqueness (**A**), Attention regulation (**B**), Emotion awareness (**C**), and Noticing (**D**) showed stronger placebo effects with a sham-personalised machine than those in the control group. Shaded regions denote 95% confidence intervals and correlations are between the trait and the pain ratings in each group.

## Expectations

Participants rated 'how effective [they] expect the machine to be in reducing [their] pain' on a scale from 0 (Not at all) to 10 (Completely). They rated their expectations twice: at the introduction of the machine, and after the conditioning.

## Side effect suggestion and assessment

To induce side effects, the experimenter suggested that approximately 10% of people using the machine may experience transient side effects: itchiness, dizziness, or muscle tremors. At the end of the pain task, they rated the experienced side effects from the machine using the modified General Assessment of Side Effects (*Rief et al., 2011*). We predicted that participants in the personalised group would show fewer side effects, because the treatment would be more personalised and less likely to cause adverse effects.

## Sample size and analyses

Analyses were similar across the two studies.

We had two hypotheses. First, we expected participants in the tailored placebo group to show a greater reduction in pain ratings than those in the control group when using the machine. We used mixed-effects linear regression (package *nlme*, R version 4.2.1) separately testing the main outcomes of pain intensity and unpleasantness given the condition (tailored or control), placebo machine state (on or off), and their interaction. We used a random intercept for each participant.

Second, we expected participants to show fewer side effects with the tailored placebo than the standard one. We ran a Poisson regression to compare the total number of reported side effects between the groups. We used a Type I error rate of .05, directional tests, and no family-wise error control.

As exploratory analyses, we also tested whether expectations and personality characteristics moderated the magnitude of placebo effects. Due to high rates of suspicion and exclusion of participants when using the pre-registered measure of expectations during pilot testing, we deviated from our pre-registered measure and instead used a single expectation rating. With this sample size and number of trials per participant, we had nearly 100% power to detect the medium behavioural effects (standardised $\beta$=0.5) found in our feasibility trial.

## Results

### Feasibility study

Participants receiving a placebo they thought was personalised reported nearly twice the reduction in pain intensity (38%; standardised $\beta$=−0.50 [−1.08, 0.08], p=.044) and unpleasantness (41%; $\beta$=−0.52 [−1.04, −0.001], p=.025, *Figures 4 and 5*) as those in the control group (19% and 27%, respectively).

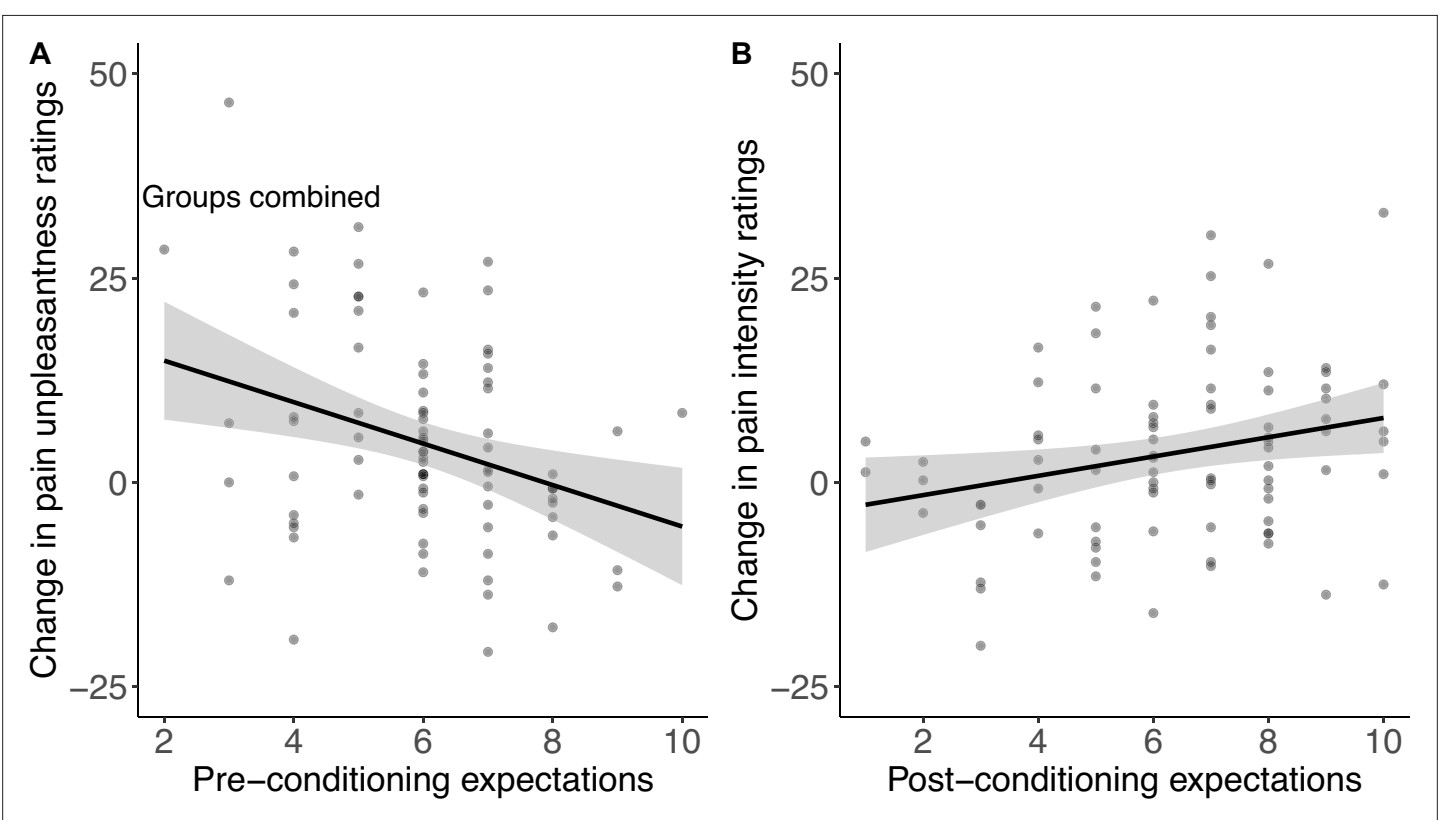

**Figure 9.** Expectations as a predictor of placebo effects with groups combined (N=84). Dots show individual scores and shaded regions denote 95% confidence intervals.

### Confirmatory study

#### Pre-registered findings: Pain ratings and side effects

Consistent with our predictions, participants in the personalised group showed stronger placebo effects than those in the control group on pain intensity (standardised $\beta=-0.20$ [–0.36, –0.04], p=.013) and unpleasantness ($\beta=-0.24$ [–0.41, –0.08], p=.003, *Figures 6 and 7*). Participants receiving a machine that they thought was personalised reported an average reduction of 5.8 points in pain intensity (11% from baseline) and a 7.3-point reduction in unpleasantness (16%), compared to the control group decrease of 1.4 points (3%) for both. The ratings of pain intensity and unpleasantness on each trial correlated nearly perfectly ($r(678)=.91$ [.89, .92], p<.001); we therefore focus on pain intensity ratings, but all effects were also found in pain unpleasantness (see Appendix 1). Several participants in both groups also reported increases in pain ratings from using the machine.

Finally, participants in both groups showed similarly low rates of side effects when using the placebo machine ($\beta_{group}=0.31$, p=.56).

#### Exploratory findings: Individual-level moderators of the placebo effect

Several personality traits moderated the personalisation placebo effects. Need for uniqueness moderated the increase in placebo analgesia in the personalised group ($\beta_{interaction}=-0.02$ [–0.03, –0.003], p=.014; *Figure 8A*). Participants with a greater need for uniqueness benefitted more from the sham personalised placebo than those in the control group.

Interoceptive awareness, measured by the Multidimensional Assessment of Interoceptive Awareness, showed a similar pattern. Three of the eight subscales of this measure drove the effects: emotion awareness (e.g., 'I notice how my body changes when I am angry'; standardised $\beta=-0.20$ [–0.35, –0.05]), attention regulation (e.g., 'I can return awareness to my body if I am distracted'; $\beta=-0.18$ [–0.34, –0.01]), and noticing (e.g., 'I notice when I am uncomfortable in my body'; $\beta=-0.17$ [–0.33, –0.01]; *Figure 8B–D*). The body listening subscale (e.g., 'When I am upset, I take time to explore how my body feels') only moderated effects on pain unpleasantness ($\beta=-0.17$ [–0.31, –0.03], see Appendix 1) but not intensity.

Other personality traits also moderated the effect on either unpleasantness (openness to experience; $\beta=-0.04$ [–0.07, –0.01]) or intensity (conscientiousness; $\beta=-0.03$ [0.003, 0.05]). Appendix 1 includes the statistics for all other personality moderators measured (*Appendix 1—tables 1 and 2*) as well as correlations between them (*Appendix 1—figure 2*). Sex did not moderate the placebo effects of personalisation ($\beta=-0.07$ [–1.09, 1.22], p=.91).

#### Exploratory findings: Expectations

Expectations about the machine's perceived effectiveness were moderate in both groups before ($M_{control}=6.1$ out of 10 (SD = 1.6), $M_{personalised}=5.9$ (1.6)) and after conditioning ($M_{control}=6.0$ (2.3), $M_{control}=6.7$ (2.2)). There was no difference between the personalised and the control conditions ($\beta_{group}=-0.30$ [–0.13, 0.73], p=.17). When combined across groups, higher pre-conditioning expectations correlated with smaller effects on pain unpleasantness ($r(82)=-.30$ [–.49, –.10], p=.005); higher post-conditioning expectations showed the opposite effect and correlated positively with stronger effects on pain intensity ($r(82)=.25$ [.04, .44], p=.021, *Figure 9*). In other words, people who expected the machine to work better *before* conditioning reported lower pain unpleasantness, while those who expected the machine to work better *after* conditioning reported lower pain intensity.

## Discussion

With interest in precision medicine and personalisation on the rise (*ANA, 2019*; *Joshua, 2019*), understanding how contextual factors influence the perceived effectiveness of targeted treatments can impact research and delivery. In a feasibility study and a pre-registered double-blind experiment, we found that completing a sham biological personalisation process led to greater placebo analgesia. In the feasibility study, participants experienced double the reduction in pain intensity when receiving treatment from a machine presented as personalised; we found similar but smaller effects in the confirmatory study. Thus, participants that received a machine framed as personalised to their genetics perceived it to be more effective in reducing their pain. Our findings provide some of the first evidence for this novel placebo effect and suggest its further study in clinical contexts, echoing

experts in the field (*Haga et al., 2009*). The results also support the need for more consistent use of blinding, inactive control groups, and randomisation, especially for pivotal trials determining FDA approval of precision drugs. Indeed, only half of the FDA-approved precision treatments in recent years were based on double- or single-blinded pivotal trials, and only 38% of all pivotal trials used a placebo comparator (*Pregelj et al., 2018*). Although precision treatments are often developed for difficult-to-study diseases, their potential to elicit stronger placebo effects calls for more robust research designs.

Better control over placebo effects in precision medicine may become especially important given the future trend of the field to use increasingly complex technologies such as brain scanning and artificial intelligence (*Ahmed et al., 2020*; *da Silva Castanheira et al., 2021*) for more extensive personalisation. Depending on the disease, targeted treatments may soon be adjusted to dozens of genetic, neural, and physiological biomarkers instead of only a few genetic markers. Such personalisation may magnify the focus on individuality, boost treatment complexity, and increase patient–practitioner interaction—likely increasing the placebo effects in the process. Preferentially using blind, randomised, and placebo-controlled trial designs can help successfully isolate the active treatment effects in such a context.

Curiously, we found that the placebo effects of personalisation may also potentially be 'personal': some participants may benefit from them more based on their personality traits. Participants high in need for uniqueness—the desire to be seen as different from others—responded strongest to the sham personalised machine, yet less so to the one in the control group. Other personality traits including attentiveness to bodily sensations (emotion awareness, attention regulation, and noticing physical sensations) as well as openness to experience also moderated the effect, in line with recent findings and the general hope for eventual personalisation of the treatment context (*Enck et al., 2013*; *Geers et al., 2006*; *Vachon-Presseau et al., 2018*). Indeed, some of the same traits (*Vachon-Presseau et al., 2018*) and general attention to symptoms (*Geers et al., 2006*) predicted increased placebo effects in other studies; our findings tentatively suggest that these traits may also amplify the specific placebo effects due to personalisation. Future studies may explore which complex personality profiles benefit the most from these placebo effects and through which mechanisms.

Several methodological strengths increased the validity of our results. We used a two-step approach of first testing the effectiveness of the deceptive procedure in a feasibility study and then confirming our findings in a pre-registered experiment; the results are thus more likely to replicate than a single study. Using the elaborate deception procedure may have also helped reduce participant suspicion (*Olson and Raz, 2021c*) and increase the reliability of their pain ratings. Only 12% of the participants suspected the placebo effect and none guessed the purpose of the experiment in the confirmatory study, despite many participants having graduate training in biology, genetics, or psychology. This is in line with previous studies on complex deception using intentionally elaborate placebos (*Olson et al., 2016*; *Olson et al., 2020*; *Olson et al., 2023*).

Finally, we imitated parts of the personalisation process such as the medical setting (*Figure 1*), therapeutic interactions (e.g., the explanation of the genetic results), and the level of complex testing (i.e., multiple tests) to somewhat increase generalisability. Together, these elements strengthened our conclusion that contextual factors may potentially play a role in increasing the placebo effect of precision treatments.

The main limitations of the study are its focus on healthy participants, the use of an inactive treatment, a sample with imbalanced genders, and the focus on subjective outcomes. Together, these factors restricted the generalisation of our findings to clinical settings. Our effect was also small; the 11% reduction in pain intensity and 16% reduction in unpleasantness reached the lower threshold of minimal clinical significance of pain reduction (10 to 20%) suggested by guidelines (*Dworkin et al., 2009*). Nevertheless, testing placebo effects with experimental pain may have led to a conservative estimate of the placebo effect and may not map directly onto the clinical experience of chronic pain. Patients differ from healthy participants on many characteristics, including their motivation to get better (*National Cancer Institute, 2021*), the mechanisms through which they experience placebo effects (short- or long-term; *Vase et al., 2005*), and the methods of assessing pain ratings (immediate versus retrospective). Our effect sizes were similar to that of paracetamol (*Jürgens et al., 2014*) and morphine (*Koppert et al., 1999*) on thermal pain, suggesting the potential for clinical significance if tested in patients. Future studies could build on our proof-of-concept findings and explore whether

these placebo effects apply to clinical populations who receive real personalised treatments focused on more objective measures. These additional investigations will help determine the clinical significance of placebo effects due to personalisation for active treatments.

Finding a mixed relationship between expectations and the tailoring process also limited our understanding of the mechanism underlying our effects. Post-conditioning expectations predicted a greater reduction in pain ratings, suggesting that conditioning is crucial for inducing placebo effects in pain, as has been demonstrated in previous studies (*Colloca et al., 2020*). However, there were no expectancy differences between the groups: both were moderate after the conditioning. The mechanism behind the placebo effect of personalisation may thus rely on an interaction with additional elements that need to be explored, such as increases in mood from receiving a personalised treatment. It is also possible that the more complex mechanism is responsible for the general lack of placebo effects in the control group, but not in the experimental group.

## Clinical implications

If the studies in clinical contexts with real treatments find a similar or larger placebo effect due to personalisation, clinicians may be able to optimise it when delivering treatments. Precision drug dosing is set to become more available to the general public by potentially targeting a broader range of diseases (*Rybak et al., 2020*); physicians may be able to enhance this placebo effect by improving therapeutic communication. For example, they could describe in detail how patients' biological variability would be used to personalise the treatment or drug dose, or they could highlight the general complexity of the personalisation procedure. Physicians could also simply emphasise the likely increase in intervention effectiveness due to its personalisation.

Outside of personalised treatments, physicians could still harness the allure of tailoring. A lot of medicine is already personalised to various metrics even before factoring in genetic testing; focusing patients' attention on that fact and *how* it is personalised to their tests or biological particularities may potentially enhance the effectiveness of more typical treatments. Indeed, placebo studies demonstrate that verbally emphasising the helpfulness of drugs like morphine further increases their effect (*Benedetti et al., 2003*). One could take a similar approach to emphasise the existing personalisation for various treatments. Overall, there are many opportunities to harness contextual factors of personalisation and patient characteristics if these are effective at improving treatment outcomes in clinical practice.

## Ideas and speculation

If confirmed in clinical settings, our findings may have implications beyond the field of precision medicine and healthcare. Individual tailoring is increasingly becoming the focus of consumer products and experiences; a large marketing organisation recently declared 'personalisation' as the word of the year (*ANA, 2019*). This may be especially true for genetics-based tailoring, likely due to the growing accessibility of testing and the general hype around genetics (*Sabatello et al., 2021*). Various companies now sell personalised diets based on nutrigenomics or personalised exercise plans based on sportomics; others promise personalised learning approaches based on behavioural genetics, to name a few. However, several of these fields are in their early stages (*Guest et al., 2019*; *Sellami et al., 2021*) and it remains unclear what the effectiveness of some such tailored approaches may be (*Janssens et al., 2008*). Our results raise the possibility that placebo effects involved in personalisation may play a relevant role in the context of the growing interest in precision medicine. In this study, we show that the personalisation process was strong enough to influence the perception of thermal pain stimulations. These effects could be potentially even more pronounced in clinical trials and medical contexts, for conditions with both objective and subjective symptoms that are amenable to placebo effects (*Wampold et al., 2005*), or for complex interventions such as diet change.

## Conclusion

We suggest a new avenue of clinical research to extend the effects of placebo personalisation to specific treatments, determine their mechanisms of action, and explore the optimisation of contextual factors in their delivery. Some interventions known to be susceptible to placebos (e.g. immunotherapy) may be more amenable to context optimisation than others (e.g., Alzheimer's therapy; *Benedetti et al., 2005*); patients from more individualistic cultures and possessing specific personality

traits may benefit from enhanced tailoring while others may be hindered by it. Initiatives like the United States' 'All of Us Program' and the UK's Biobank are collecting millions of data points on biomarkers of disease in a move towards routinely personalised healthcare. We show that contextual factors may be a hidden element to understand and harness in this new era of medicine.

## Acknowledgements

The authors thank Mira Kaedbey, Naz Alpdogan, and Holly Bowman for assisting with data collection; Alain Al Bikaii for feedback on the manuscript; as well as Samuel Veissière, Michael Lifshitz, Jason da Silva Castanheira, and the Langer Mindfulness Lab for helpful suggestions. We also thank Ekaterina Rossokhata for inspiring interest in this research.

This study was funded through the Social Sciences and Humanities Research Council (SSHRC, PT 93188), and Genome Québec (PT 95747). JO acknowledges funding from the Fonds de Recherche du Quebec—Santé (FRQS). DS acknowledges funding from the Fonds de Recherche du Quebec Nature et Téchnologie (FRQNT).

## Additional information

### Funding

| Funder | Grant reference number | Author |
| --- | --- | --- |
| Social Sciences and Humanities Research Council of Canada | PT 93188 | Mathieu Roy Jay A Olson |
| Genome Quebec | PT 95747 | Mathieu Roy Jay A Olson |

The funders had no role in study design, data collection and interpretation, or the decision to submit the work for publication.

### Author contributions

Dasha A Sandra, Conceptualization, Data curation, Formal analysis, Funding acquisition, Validation, Investigation, Visualization, Methodology, Writing – original draft, Project administration, Writing – review and editing; Jay A Olson, Conceptualization, Supervision, Funding acquisition, Methodology, Project administration, Writing – review and editing; Ellen J Langer, Conceptualization, Supervision, Methodology, Writing – review and editing; Mathieu Roy, Conceptualization, Supervision, Funding acquisition, Methodology, Writing – review and editing

### Author ORCIDs

Dasha A Sandra ⓘ http://orcid.org/0000-0001-9930-2807
Jay A Olson ⓘ http://orcid.org/0000-0002-1161-5209
Mathieu Roy ⓘ http://orcid.org/0000-0002-3335-445X

### Ethics

The study was approved by the McGill University Research Ethics Board II (#45-0619). All participants consented to participate and to their results to be a part of group analysis which were published upon completion of the study.

### Decision letter and Author response

Decision letter https://doi.org/10.7554/eLife.84691.sa1
Author response https://doi.org/10.7554/eLife.84691.sa2

## Additional files

### Supplementary files
• MDAR checklist

## Data availability

All data is freely available at Open Science Framework (https://osf.io/6j7z5/).

The following dataset was generated:

| Author(s) | Year | Dataset title | Dataset URL | Database and Identifier |
|---|---|---|---|---|
| Sandra DA, Olson JA, Langer EJ, Roy M | 2023 | Pain ratings with and without two types of a placebo machine | https://osf.io/6j7z5/ | Open Science Framework, 6j7z5 |

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

# Appendix 1

## Measures

### Need for Uniqueness (NUS)

The NUS is a 32-item self-report measure assessing a person's motivation to appear different or unique (*Snyder and Fromkin, 1977*). Participants rate characteristics like "Feeling 'different' in a crowd of people makes me feel uncomfortable" on a scale of 1 (Strongly disagree) to 5 (Strongly agree). It ranges between 32 and 160, and has a high internal reliability (Cronbach's $\alpha$=.84).

### Multidimensional Assessment of Interoceptive Awareness (MAIA)

To measure interoceptive awareness, or attention to bodily sensations, we used the MAIA scale (*Mehling et al., 2012*). It includes 32 questions on 8 different aspects of interoceptive awareness, such as noticing one's sensations ('I notice when I am uncomfortable in my body'), awareness of bodily sensations and emotional states ('When something is wrong in my life, I can feel it in my body'), and regulating one's attention to sensations ('I can return awareness to my body if I am distracted'). The scale ranges from 0 to 160 in total, but each subscale can have its own score. Each subscale score is computed as the mean of all questions included in that subscale. Reliability of each of these varies from adequate to good ($\alpha$=.66 to .82).

### Fear of Pain Questionnaire-III (FPQ-III)

Pain anxiety and desire for pain relief may predict the magnitude of the experienced placebo analgesia (*Wager, 2005*). The FPQ-III is a 30-item self-report measure assessing fear in response to painful stimuli (*McNeil and Rainwater, 1998*). Participants rate fear of painful experiences such as 'Breaking your arm' on a scale of 1 (Not at all) to 5 (Extreme), with scores ranging from 1 to 150. Subscales have excellent internal consistency ($\alpha$ ranging from .88 to .92).

### Pain Catastrophising Questionnaire (PCS)

The PCS is a 13-item self-report measure assessing the trait for catastrophising thoughts related to pain (*Sullivan et al., 1995*). Participants rate thoughts and feelings such as 'I feel I can't go on' about the experience of pain on a scale of 1 (Not at all) to 4 (All the time). The score range is between 13 and 52; the higher the score, the more catastrophising thoughts are present. This questionnaire has excellent internal consistency ($\alpha$=.93).

### Big Five Inventory (BFI)

The BFI is a 44-item self-report measure assessing five broad personality traits: openness to experience, conscientiousness, extraversion, neuroticism, and agreeableness (*John et al., 1991*; *John and Srivastava, 1999*). Participants rate characteristics like 'I am someone who is talkative' on a scale of 1 (Disagree strongly) to 5 (Agree strongly). It has good internal reliability ($\alpha$=.83); each trait has a separate score, summed across its respective subscale items.

Pain ratings during conditioning

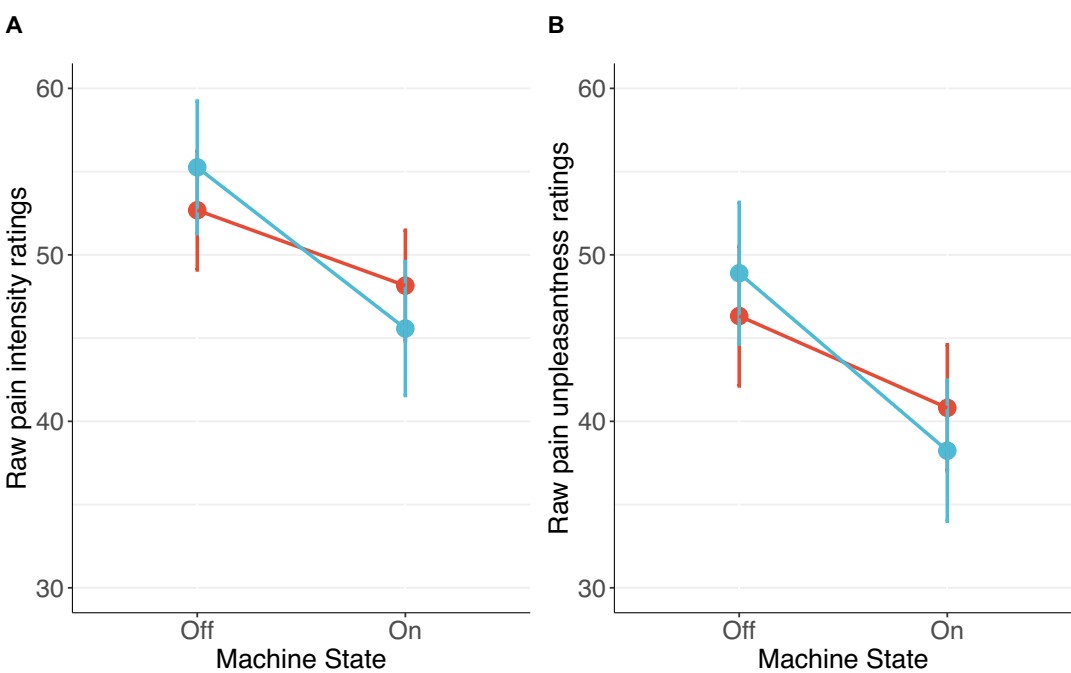

**Appendix 1—figure 1.** The differences in pain intensity and unpleasantness during the conditioning phase of the confirmatory study. Dots show means and error bars show 95% confidence intervals.

## Personality trait moderators

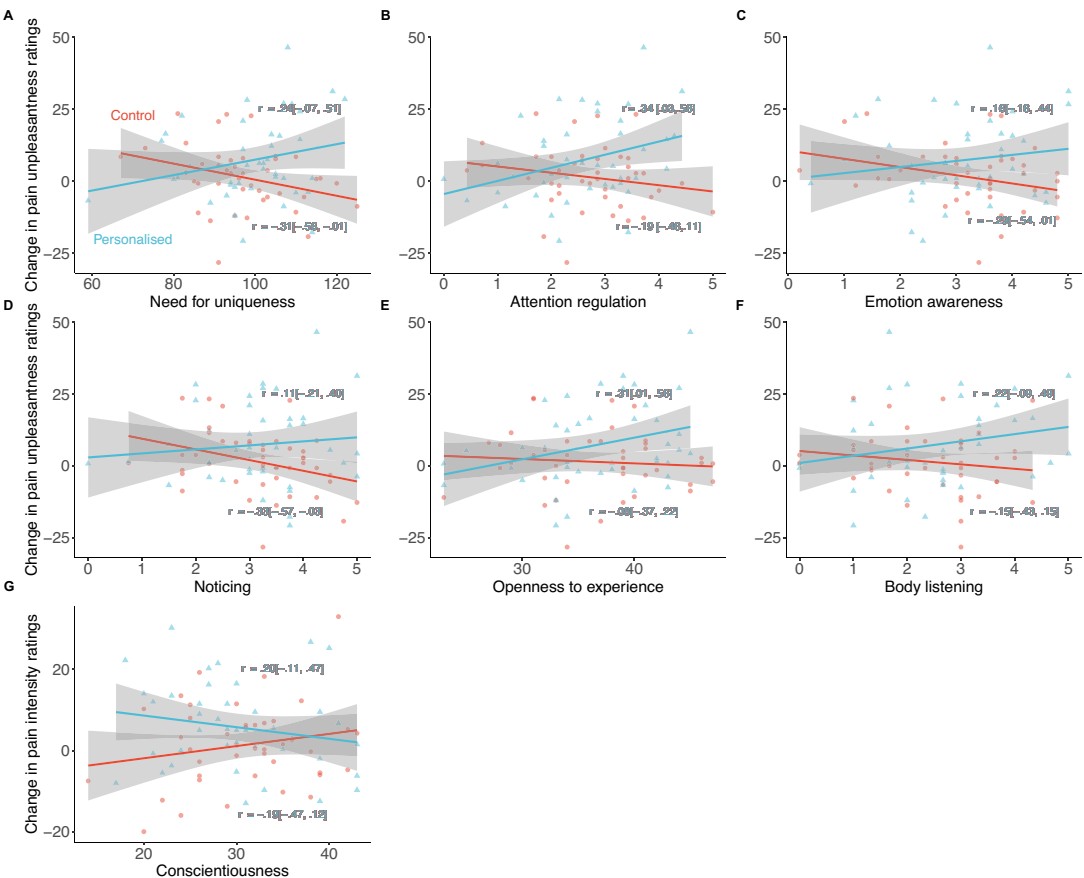

**Appendix 1—figure 2.** Personality traits that significantly moderated the placebo effects of personalisation (N=85). Shaded regions show 95% confidence intervals, equations represent proportion of variance explained by each group.

**Appendix 1—table 1.** Regression results of all personality predictors of increased placebo effects on pain intensity.

We only tested interactions to reduce the probability of Type I errors; all tests were exploratory. Significant interactions (change in pain ratings × personality trait; two-tailed *p* <.05) are bolded.

| Personality trait | Predictor | Standardised *β* | SE | df | t | p |
|---|---|---|---|---|---|---|
| Attention regulation | (Intercept) | 0.034 | 0.411 | 591 | 0.084 | .933 |
| | Condition | –0.032 | 0.549 | 81 | –0.059 | .953 |
| | Machine | –0.167 | 0.175 | 591 | –0.950 | .342 |
| | Attention regulation | 0.020 | 0.143 | 81 | 0.138 | .891 |
| | **Interaction** | **–0.177** | **0.083** | **591** | **–2.142** | **.033** |
| Noticing | (Intercept) | –0.158 | 0.46 | 591 | –0.343 | .732 |
| | Condition | 0.058 | 0.623 | 81 | 0.093 | .926 |
| | Machine | –0.414 | 0.198 | 591 | –2.094 | .037 |
| | Noticing | 0.077 | 0.139 | 81 | 0.554 | .581 |
| | **Interaction** | **–0.167** | **0.081** | **591** | **–2.065** | **.039** |
| Not-worrying | (Intercept) | 0.271 | 0.365 | 591 | 0.743 | .458 |

*Appendix 1—table 1 Continued on next page*

*Appendix 1—table 1 Continued*

| Personality trait | Predictor | Standardised β | SE | df | t | p |
|---|---|---:|---:|---:|---:|---:|
| | Condition | −0.121 | 0.493 | 81 | −0.244 | .807 |
| | Machine | 0.047 | 0.157 | 591 | 0.297 | .766 |
| | Not-worrying | −0.074 | 0.138 | 81 | −0.537 | .593 |
| | Interaction | 0.080 | 0.082 | 591 | 0.981 | .327 |
| Self-regulation | (Intercept) | −0.267 | 0.401 | 591 | −0.666 | .506 |
| | Condition | 0.272 | 0.541 | 81 | 0.502 | .617 |
| | Machine | −0.095 | 0.173 | 591 | −0.547 | .584 |
| | Self-regulation | 0.132 | 0.140 | 81 | 0.940 | .350 |
| | Interaction | −0.077 | 0.082 | 591 | −0.943 | .346 |
| Emotion awareness | (Intercept) | −0.333 | 0.413 | 591 | −0.806 | .421 |
| | Condition | 0.693 | 0.591 | 81 | 1.172 | .244 |
| | Machine | −0.435 | 0.179 | 591 | −2.426 | .016 |
| | Emotion awareness | 0.130 | 0.121 | 81 | 1.078 | .284 |
| | **Interaction** | **−0.203** | **0.077** | **591** | **−2.643** | **.008** |
| Not-distracting | (Intercept) | 0.739 | 0.335 | 591 | 2.207 | .028 |
| | Condition | −1.030 | 0.522 | 81 | −1.974 | .052 |
| | Machine | −0.205 | 0.147 | 591 | −1.391 | .165 |
| | Not-distracting | −0.312 | 0.147 | 81 | −2.119 | .037 |
| | Interaction | −0.134 | 0.100 | 591 | −1.350 | .177 |
| Trusting | (Intercept) | 0.299 | 0.423 | 591 | 0.707 | .480 |
| | Condition | −0.945 | 0.572 | 81 | −1.651 | .103 |
| | Machine | 0.013 | 0.185 | 591 | 0.069 | .945 |
| | Trusting | −0.065 | 0.123 | 81 | −0.529 | .598 |
| | Interaction | −0.037 | 0.073 | 591 | −0.503 | .615 |
| Body listening | (Intercept) | −0.296 | 0.327 | 591 | −0.904 | .366 |
| | Condition | 0.155 | 0.447 | 81 | 0.348 | .729 |
| | Machine | −0.106 | 0.142 | 591 | −0.749 | .454 |
| | Body listening | 0.157 | 0.123 | 81 | 1.276 | .206 |
| | Interaction | −0.068 | 0.071 | 591 | −0.947 | .344 |
| Openness to experience | (Intercept) | 0.387 | 0.839 | 591 | 0.462 | .644 |
| | Condition | 0.515 | 1.205 | 81 | 0.427 | .671 |
| | Machine | −0.129 | 0.362 | 591 | −0.356 | .722 |
| | Openness to experience | −0.008 | 0.023 | 81 | −0.360 | .720 |
| | Interaction | −0.003 | 0.014 | 591 | −0.226 | .821 |
| Conscientiousness | (Intercept) | 0.062 | 0.646 | 591 | 0.097 | .923 |
| | Condition | −0.251 | 0.877 | 81 | −0.286 | .776 |
| | Machine | 0.371 | 0.277 | 591 | 1.336 | .182 |

*Appendix 1—table 1 Continued on next page*

*Appendix 1—table 1 Continued*

| Personality trait | Predictor | Standardised β | SE | df | t | p |
|---|---|---|---|---|---|---|
| | Conscientiousness | 0.001 | 0.020 | 81 | 0.043 | .966 |
| | **Interaction** | **0.027** | **0.012** | **591** | **2.281** | **.023** |
| Extraversion | (Intercept) | 0.625 | 0.544 | 591 | 1.148 | .251 |
| | Condition | –0.246 | 0.825 | 81 | –0.298 | .766 |
| | Machine | –0.499 | 0.233 | 591 | –2.136 | .033 |
| | Extraversion | –0.021 | 0.021 | 81 | –1.015 | .313 |
| | Interaction | –0.013 | 0.013 | 591 | –0.953 | .341 |
| Agreeableness | (Intercept) | 0.451 | 0.738 | 591 | 0.611 | .541 |
| | Condition | –0.763 | 1.416 | 81 | –0.538 | .592 |
| | Machine | 0.323 | 0.317 | 591 | 1.017 | .310 |
| | Agreeableness | –0.011 | 0.022 | 81 | –0.500 | .618 |
| | Interaction | 0.001 | 0.018 | 591 | 0.061 | .951 |
| Neuroticism | (Intercept) | 0.578 | 0.547 | 591 | 1.057 | .291 |
| | Condition | 0.093 | 0.803 | 81 | 0.115 | .908 |
| | Machine | –0.371 | 0.236 | 591 | –1.573 | .116 |
| | Neuroticism | –0.020 | 0.021 | 81 | –0.920 | .361 |
| | Interaction | –0.008 | 0.013 | 591 | –0.594 | .553 |
| Fear of pain | (Intercept) | 0.296 | 0.735 | 591 | 0.403 | .687 |
| | Condition | –1.179 | 1.070 | 81 | –1.102 | .274 |
| | Machine | –0.493 | 0.318 | 591 | –1.553 | .121 |
| | Fear of pain | –0.002 | 0.009 | 81 | –0.283 | .778 |
| | Interaction | –0.007 | 0.005 | 591 | –1.211 | .226 |
| Pain catastrophising | (Intercept) | –0.221 | 0.282 | 591 | –0.786 | .432 |
| | Condition | 0.268 | 0.449 | 81 | 0.596 | .553 |
| | Machine | –0.095 | 0.122 | 591 | –0.776 | .438 |
| | Pain catastrophising | 0.014 | 0.012 | 81 | 1.246 | .216 |
| | Interaction | –0.009 | 0.008 | 591 | –1.104 | .270 |

**Appendix 1—table 2.** Regression results of all personality predictors of increased placebo effects on pain unpleasantness.
Significant interactions (change in pain ratings × personality trait; two-tailed *p* <.05) are bolded.

| Personality trait | Predictor | β | SE | df | t | p |
|---|---|---|---|---|---|---|
| Attention regulation | (Intercept) | 0.036 | 0.413 | 591 | 0.086 | .931 |
| | Condition | 0.052 | 0.552 | 81 | 0.094 | .925 |
| | Machine | –0.302 | 0.176 | 591 | –1.710 | .088 |
| | Attention regulation | –0.010 | 0.144 | 81 | –0.067 | .947 |
| | **Interaction** | **–0.279** | **0.083** | **591** | **–3.349** | **.001** |
| Noticing | (Intercept) | –0.416 | 0.460 | 591 | –0.905 | .366 |
| | Condition | 0.445 | 0.623 | 81 | 0.714 | .477 |
| | Machine | –0.545 | 0.200 | 591 | –2.724 | .007 |

*Appendix 1—table 2 Continued on next page*

Appendix 1—table 2 Continued

| Personality trait | Predictor | β | SE | df | t | p |
|---|---|---|---|---|---|---|
|  | Noticing | 0.134 | 0.139 | 81 | 0.967 | .336 |
|  | **Interaction** | **−0.212** | **0.082** | **591** | **−2.590** | **.010** |
| Not-worrying | (Intercept) | 0.357 | 0.362 | 591 | 0.988 | .324 |
|  | Condition | 0.309 | 0.488 | 81 | 0.632 | .529 |
|  | Machine | 0.008 | 0.159 | 591 | 0.052 | .958 |
|  | Not-worrying | −0.144 | 0.137 | 81 | −1.049 | .297 |
|  | Interaction | 0.078 | 0.083 | 591 | 0.944 | .346 |
| Self-regulation | (Intercept) | −0.343 | 0.403 | 591 | −0.851 | .395 |
|  | Condition | 0.443 | 0.544 | 81 | 0.814 | .418 |
|  | Machine | −0.215 | 0.175 | 591 | −1.227 | .220 |
|  | Self-regulation | 0.132 | 0.141 | 81 | 0.932 | .354 |
|  | Interaction | −0.133 | 0.083 | 591 | −1.613 | .107 |
| Emotion awareness | (Intercept) | −0.463 | 0.417 | 591 | −1.111 | .267 |
|  | Condition | 0.679 | 0.596 | 81 | 1.139 | .258 |
|  | Machine | −0.440 | 0.182 | 591 | −2.419 | .016 |
|  | Emotion awareness | 0.146 | 0.122 | 81 | 1.198 | .234 |
|  | **Interaction** | **−0.206** | **0.078** | **591** | **−2.644** | **.008** |
| Not-distracting | (Intercept) | 0.632 | 0.338 | 591 | 1.869 | .062 |
|  | Condition | −0.961 | 0.527 | 81 | −1.824 | .072 |
|  | Machine | −0.207 | 0.149 | 591 | −1.389 | .165 |
|  | Not-distracting | −0.296 | 0.149 | 81 | −1.992 | .050 |
|  | Interaction | −0.184 | 0.101 | 591 | −1.823 | .069 |
| Trusting | (Intercept) | 0.206 | 0.430 | 591 | 0.479 | .632 |
|  | Condition | −0.580 | 0.582 | 81 | −0.997 | .322 |
|  | Machine | −0.065 | 0.188 | 591 | −0.346 | .730 |
|  | Trusting | −0.061 | 0.125 | 81 | −0.485 | .629 |
|  | Interaction | −0.008 | 0.074 | 591 | −0.108 | .914 |
| Body listening | (Intercept) | −0.427 | 0.325 | 591 | −1.315 | .189 |
|  | Condition | 0.141 | 0.444 | 81 | 0.317 | .752 |
|  | Machine | −0.213 | 0.143 | 591 | −1.484 | .138 |
|  | Body listening | 0.180 | 0.122 | 81 | 1.471 | .145 |
|  | **Interaction** | **−0.168** | **0.072** | **591** | **−2.331** | **.020** |
| Openness to experience | (Intercept) | 0.374 | 0.845 | 591 | 0.442 | .658 |
|  | Condition | 0.055 | 1.214 | 81 | 0.045 | .964 |
|  | Machine | −0.287 | 0.363 | 591 | −0.790 | .430 |
|  | Openness to experience | −0.010 | 0.023 | 81 | −0.433 | .666 |
|  | **Interaction** | **−0.037** | **0.014** | **591** | **−2.655** | **.008** |
| Conscientiousness | (Intercept) | −0.042 | 0.654 | 591 | −0.064 | .949 |

Appendix 1—table 2 Continued on next page

*Appendix 1—table 2 Continued*

| Personality trait | Predictor | β | SE | df | t | p |
|---|---|---|---|---|---|---|
| | Condition | 0.409 | 0.888 | 81 | 0.460 | .647 |
| | Machine | 0.136 | 0.282 | 591 | 0.482 | .630 |
| | Conscientiousness | 0.002 | 0.021 | 81 | 0.081 | .936 |
| | Interaction | 0.020 | 0.012 | 591 | 1.608 | .108 |
| Extraversion | (Intercept) | 0.541 | 0.548 | 591 | 0.987 | .324 |
| | Condition | –0.531 | 0.831 | 81 | –0.639 | .525 |
| | Machine | –0.380 | 0.237 | 591 | –1.603 | .110 |
| | Extraversion | –0.021 | 0.021 | 81 | –1.004 | .318 |
| | Interaction | –0.018 | 0.014 | 591 | –1.326 | .185 |
| Agreeableness | (Intercept) | 0.364 | 0.743 | 591 | 0.489 | .625 |
| | Condition | –0.723 | 1.427 | 81 | –0.507 | .614 |
| | Machine | 0.250 | 0.322 | 591 | 0.777 | .438 |
| | Agreeableness | –0.011 | 0.022 | 81 | –0.484 | .629 |
| | Interaction | 0.007 | 0.018 | 591 | 0.371 | .710 |
| Neuroticism | (Intercept) | 0.029 | 0.554 | 591 | 0.053 | .958 |
| | Condition | –0.016 | 0.814 | 81 | –0.020 | .984 |
| | Machine | –0.338 | 0.239 | 591 | –1.414 | .158 |
| | Neuroticism | –0.001 | 0.022 | 81 | –0.037 | .970 |
| | Interaction | –0.015 | 0.014 | 591 | –1.137 | .256 |
| Fear of pain | (Intercept) | –0.156 | 0.742 | 591 | –0.210 | .834 |
| | Condition | –0.405 | 1.080 | 81 | –0.375 | .709 |
| | Machine | –0.441 | 0.322 | 591 | –1.370 | .171 |
| | Fear of pain | 0.002 | 0.009 | 81 | 0.231 | .818 |
| | Interaction | –0.004 | 0.006 | 591 | –0.671 | .502 |
| Pain catastrophising | (Intercept) | –0.404 | 0.278 | 591 | –1.452 | .147 |
| | Condition | 0.064 | 0.444 | 81 | 0.143 | .886 |
| | Machine | –0.098 | 0.124 | 591 | –0.796 | .426 |
| | Pain catastrophising | 0.019 | 0.011 | 81 | 1.678 | .097 |
| | Interaction | –0.010 | 0.008 | 591 | –1.172 | .242 |

|  | MAIA | Noticing | Not–distracting | Not–worrying | Attention regulation | Emotion awareness | Self regulation | Body listening | Trusting | Agreeableness | Conscientiousness | Extraversion | Neuroticism | Openness to experience | Need for uniqueness | Fear of pain | Pain catastrophising |
|---|---|---|---|---|---|---|---|---|---|---|---|---|---|---|---|---|---|
| MAIA |  | 0.73 | 0.12 | 0.16 | 0.83 | 0.73 | 0.8 | 0.78 | 0.56 | 0.19 | 0.39 | 0.2 | −0.34 | 0.33 | 0.26 | −0.02 | −0.02 |
| Noticing | 0.73 |  | 0.11 | 0.04 | 0.54 | 0.52 | 0.43 | 0.56 | 0.25 | 0.09 | 0.24 | 0.2 | −0.07 | 0.3 | 0.26 | 0.06 | 0.11 |
| Not–distracting | 0.12 | 0.11 |  | −0.27 | −0.03 | 0.05 | −0.05 | 0.03 | 0.14 | 0.17 | 0.13 | −0.02 | 0.11 | −0.15 | −0.14 | 0.19 | 0.15 |
| Not–worrying | 0.16 | 0.04 | −0.27 |  | 0.21 | −0.1 | 0.05 | −0.09 | 0.04 | 0.24 | 0.22 | 0.05 | −0.27 | −0.02 | 0.09 | −0.22 | −0.58 |
| Attention regulation | 0.83 | 0.54 | −0.03 | 0.21 |  | 0.44 | 0.64 | 0.56 | 0.33 | 0 | 0.24 | 0.08 | −0.28 | 0.39 | 0.39 | −0.11 | −0.11 |
| Emotion awareness | 0.73 | 0.52 | 0.05 | −0.1 | 0.44 |  | 0.51 | 0.61 | 0.25 | 0.09 | 0.2 | 0.14 | −0.08 | 0.22 | 0.11 | 0.08 | 0.26 |
| Self regulation | 0.8 | 0.43 | −0.05 | 0.05 | 0.64 | 0.51 |  | 0.65 | 0.51 | 0.19 | 0.33 | 0.15 | −0.42 | 0.26 | 0.14 | −0.1 | −0.06 |
| Body listening | 0.78 | 0.56 | 0.03 | −0.09 | 0.56 | 0.61 | 0.65 |  | 0.36 | 0.09 | 0.28 | 0.14 | −0.16 | 0.34 | 0.12 | 0.03 | 0.11 |
| Trusting | 0.56 | 0.25 | 0.14 | 0.04 | 0.33 | 0.25 | 0.51 | 0.36 |  | 0.28 | 0.38 | 0.29 | −0.5 | 0.05 | 0.02 | 0.02 | −0.08 |
| Agreeableness | 0.19 | 0.09 | 0.17 | 0.24 | 0 | 0.09 | 0.19 | 0.09 | 0.28 |  | 0.31 | 0.1 | −0.37 | −0.02 | −0.12 | −0.03 | −0.13 |
| Conscientiousness | 0.39 | 0.24 | 0.13 | 0.22 | 0.24 | 0.2 | 0.33 | 0.28 | 0.38 | 0.31 |  | 0.15 | −0.35 | −0.07 | −0.13 | −0.01 | −0.22 |
| Extraversion | 0.2 | 0.2 | −0.02 | 0.05 | 0.08 | 0.14 | 0.15 | 0.14 | 0.29 | 0.1 | 0.15 |  | −0.22 | 0.09 | 0.41 | −0.09 | −0.07 |
| Neuroticism | −0.34 | −0.07 | 0.11 | −0.27 | −0.28 | −0.08 | −0.42 | −0.16 | −0.5 | −0.37 | −0.35 | −0.22 |  | −0.06 | −0.13 | 0.14 | 0.29 |
| Openness to experience | 0.33 | 0.3 | −0.15 | −0.02 | 0.39 | 0.22 | 0.26 | 0.34 | 0.05 | −0.02 | −0.07 | 0.09 | −0.06 |  | 0.54 | −0.04 | 0.21 |
| Need for uniqueness | 0.26 | 0.26 | −0.14 | 0.09 | 0.39 | 0.11 | 0.14 | 0.12 | 0.02 | −0.12 | −0.13 | 0.41 | −0.13 | 0.54 |  | −0.1 | 0 |
| Fear of pain | −0.02 | 0.06 | 0.19 | −0.22 | −0.11 | 0.08 | −0.1 | 0.03 | 0.02 | −0.03 | −0.01 | −0.09 | 0.14 | −0.04 | −0.1 |  | 0.42 |
| Pain catastrophising | −0.02 | 0.11 | 0.15 | −0.58 | −0.11 | 0.26 | −0.06 | 0.11 | −0.08 | −0.13 | −0.22 | −0.07 | 0.29 | 0.21 | 0 | 0.42 |  |

**Appendix 1—figure 3.** Correlations between all personality traits measured as potential predictors of placebo effects of personalisation.

