## [Editor Report]

Sandra et al. assessed the effects of a personalized intervention on the placebo effect in a randomized controlled trial. The study showcases important results highlighting that psychological aspects of 'personalised' or 'precision' medicine substantially shape the treatment effects over and above the benefit of biologically/clinically/pharmacologically tailored interventions. It has to be noted that the effect sizes identified are relatively small and the outcomes are subjective, which has implications for the generalizability of the results.

---

## [Decision Letter]

**Decision letter after peer review:**

Thank you for submitting your article "Presenting a Sham Treatment as Personalised Increases its Effectiveness in a Randomised Controlled Trial" for consideration by *eLife*. Your article has been reviewed by 3 peer reviewers, including José Biurrun Manresa as the Reviewing Editor and Reviewer #1, and the evaluation has been overseen by Christian Büchel as the Senior Editor.

Essential revisions:

Reviewers have outlined several recommendations for the authors. Below please find a summarized list with the essential revisions, but do refer to the reviewers' suggestions for details:

1) Revise the terminology throughout the manuscript.

2) Add missing details about the methodology and additional data regarding calibration procedures.

3) Improve the graphical data presentation.

4) Add the statistical analysis on the existing data requested by the reviewers.

5) Rework the discussion and reassess the extent of the claims taking into account the reviewers' suggestion, particularly with regards to the magnitude of the effects and its clinical significance, and the potential confounders that are not currently discussed (sample bias, intervening variables, etc.).

*Reviewer #1:*

In this manuscript, Sandra et al. aimed at quantifying the role of the placebo effects of personalization in a randomized clinical trial.

The main strengths of the manuscript are:

– It presents data from an exploratory and a confirmatory study to test the same hypothesis.

– The study presents data from several relevant variables that appear to have been carefully collected.

The main weaknesses of the manuscript are:

– The sample is not representative of the general population and the experimental settings are not a good match for clinical settings, which hinders the generalizability of the results.

– The interpretation of the results does not consider potential implications related to individual vs group differences, or the experimental or clinical relevance of the effect sizes observed.

I believe that the authors partially succeed in their aim, given that they are able to show a group effect of personalization in the quantification of the placebo effect. I believe that the discussion would benefit from contextualizing these results in the experimental settings, and reappraising them in relation to their actual clinical relevance.

Terminology

This might sound like a minor semantic detail, but the authors state that "precision treatments benefit from contextual factors", and that "treatment effects can be boosted by personalization", and this phrasing considers the placebo effect as part of the treatment. If I might suggest a different phrasing, I would say that the outcomes of an "intervention" can be constructed as the sum of the "real" treatment effect (if any) plus the placebo effect, and personalization in the context of this study only boosts the placebo effect. Here, the word "treatment" is used with a double meaning: as the action of attending to the patient's needs (what I suggest calling the intervention), and as the active component of the therapy that is supposed to produce physiological effects through mechanisms other than the placebo effect, that is, the treatment (which could be a pain-relief drug or a procedure such as TENS). Whereas this is not wrong per se, I would argue that the manuscript would benefit from clearly differentiating between these two terms, because this study does not have an active "treatment" condition, i.e., a group in which analgesia was provided by means known to work directly at the nociceptive system level. This would also be helpful to handle ethical issues that might arise if one considers placebo as part of the treatment: under these conclusions, pseudo-scientific therapies not backed by evidence (e.g., homeopathy) could be called "effective" just by "tailoring" or "personalizing" some aspect of the treatment to specific patient traits.

In line with this, the use of the term "treatment effectiveness" might not be entirely adequate to refer to the placebo analgesia effects. A suggestion would be to qualify it as "perceived effectiveness", or to rephrase it as "Presenting a sham treatment as personalized increases the placebo effect in an RCT". Furthermore, there is no real "pain relief": as the figures show, there are changes in pain intensity ratings due to the summed effects of placebo and reduction in stimulation intensity.

Experimental design

Settings and sampling

I believe that the settings and sampling could have been improved. Even though the authors clearly state that volunteers were dropped if they asked about or mentioned the placebo effect, all participants are sampled from the McGill University community, and from the data, most are women who are also undergraduate students. Furthermore, it is fair to assume that the experiments were carried out in a laboratory at the Dept. of Psychology. In most circumstances (and for the feasibility study presented here) one would accept that an exploratory study might be carried out in these settings. However, the confirmatory study deals with a well-known psychological effect, is carried out in the Dept. of Psychology by psychologists, the volunteers are mostly sampled from psychology students, and the experiment has a large number of psychological questions related to the expectation of pain. Furthermore, the "pain relief" device supposedly tailored to genetic information looks like technology from the 70s, when they could have just used the TENS stimulator that probably looks like cutting-edge, pain relief technology.

Interestingly, the authors list many of these parts of the study as strengths, but I believe some of them are limitations. I actually find it odd that only 12% of participants having graduate training in biology, genetics, or psychology would not behave differently in this experiment (more on these below in the results subsection), and we actually have no data on how the general population would behave; the authors could have run the experiment on real medical settings with medical doctors (as opposed to psychology researchers) and a level of complexity during testing that mimics clinical routine, no more and no less (the abundance of psychological questions is particularly notorious).

Sham condition

If I followed the methodology correctly, the expected change in pain intensity in the conditioning stage of the feasibility study should be about 60 points for the feasibility study (80-20) and 40 points for the confirmatory study (60-20). The effect on pain intensity ratings is so large that participants should believe that this is nothing short of a miraculous machine (considering experimental settings, age, gender, and current health status). I suggest authors present data recorded on the observed effect in pain intensity and unpleasantness ratings during conditioning.

Following the experiment, participants proceed to the testing block, and the effect almost vanishes (an average of 5 points on pain intensity ratings, approximately 10% of the effect during the conditioning stage). I find it surprising that this fact apparently did not come up during the debriefing. Furthermore, debriefing is mentioned in the methods, but no results are reported in this regard.

Interpretation of results

I assume that the results shown in Figure 3 and 4 are the change in pain intensity and unpleasantness ratings during the testing block, for which the temperature at both the On and Off conditions is calibrated at a pain intensity of 50. I suggest reporting data from the calibration procedure in more detail, in order to compare it with the extensive knowledge available for thermal pain testing (e.g. Moloney et al., 2012).

Furthermore, in the absence of a placebo effect (or other psychological confounders derived from the experimental settings), one would expect ratings during On and Off testing to be distributed around 50 points. In this regard, I suggest authors plot not just the differences, but the actual recorded ratings (with lines linking the On and Off data for each subject, since these are within-subject repeated measurements), and analyze the results using the Off condition as covariate (Vickers, 2001).

In line with this, differences between On and Off testing are to be within the measurement error for pain intensity ratings due to thermal stimulation, with an average difference between conditions of zero (no bias), as observed in the control condition. Since the authors report a non-zero average difference (5.8/100) they attribute this to the placebo effect.

However, the settings for this experiment are quite particular: since it discusses "personalized" effects, it is important to pay attention to individual pain intensity and unpleasantness scores. From the figures, if one considers a binary scale, in which a difference of zero is no effect, any positive difference (regardless of its size) implies a placebo effect, and any negative difference (regardless of its size) implies a nocebo effect, then a significant number of participants is individually reporting nocebo effects in both conditions, i.e., they report more pain and unpleasantness during the On condition. This is surprisingly not mentioned anywhere in the text.

If, on the other hand, one is inclined to interpret these results as a group effect, the experimental/clinical relevance of the effect size must be considered, instead of just the statistical significance (Angst, 2017; de Vet et al., 2006). In this section of the Discussions, the authors should discuss the actual effect size (5.8/100) and its relation to the minimal clinically relevant effect. In my opinion, the detected group effect is too small to be considered clinically significant. The authors state "Our findings raise a question in the broad context of increasing interest in personalisation: just how big of a placebo effect is there from intervention tailoring? In this study, we show that the personalisation process was strong enough to influence the perception of thermal pain stimulations". While I agree that the personalization process somewhat influenced the perception of thermal pain thresholds at a group level, I also believe that this was only detectable with a relatively large sample, that the effect size is small (and in my opinion, not clinically significant), and that the implications of nocebo effects in individual pain ratings were not thoroughly analyzed and discussed.

References

Angst, F. (2017). The minimal clinically important difference raised the significance of outcome effects above the statistical level, with methodological implications for future studies. Journal of Clinical Epidemiology.

Moloney, N.A., Hall, T.M., Doody, C.M. (2012). Reliability of thermal quantitative sensory testing: A systematic review. Journal of Rehabilitation Research and Development 49, 191-208.

de Vet, H.C., Terwee, C.B., Ostelo, R.W., Beckerman, H., Knol, D.L., Bouter, L.M. (2006). Minimal changes in health status questionnaires: distinction between minimally detectable change and minimally important change. Health Qual Life Outcomes 4, 54.

Vickers, A.J. (2001). The use of percentage change from baseline as an outcome in a controlled trial is statistically inefficient: a simulation study. BMC Medical Research Methodology 1, 6.

*Reviewer #2:*

This pre-registered confirmatory study provides experimental evidence that the 'personalisation' of a treatment represents an important contextual factor that can enhance the placebo analgesic effect. This is shown in a well-controlled, double-blind study, where a sham analgesic treatment is either presented as "personalized" or not in 85 healthy volunteers. The authors show that volunteers in the 'personalized' group display stronger placebo analgesic responses as compared to the control group (although it should be noted that overall no placebo analgesic effect is induced in the control group, which is somewhat surprising given the conditioning procedure used). No differences in side effect profiles were observed, however, side effects were generally low in both groups, so the absence of differences between groups might be explained by a floor effect.

Interestingly, these differences in placebo analgesia depending on 'personalisation' are not paralleled by significant differences in expectancy levels between groups neither before nor after conditioning. The authors also explore the role of different personality traits that may modulate the effect of 'personalisation' with interesting findings.

Overall, the manuscript is very well written, and the rationale of the study is clear and highly relevant, and innovative in the context of the increasing attempts to individually tailor treatments, also referred to as precision medicine. The methods of the study are sound and the conclusions of the authors are adequately backed up by the data.

These proof-of-concept findings obtained in healthy volunteers may have broad implications for the design and interpretation of clinical trials as well as systematic attempts to optimize treatment effects in clinical settings in the field of pain and beyond. Next studies have to test, whether the results translate into clinical contexts and beyond placebo analgesia.

The authors should indicate the amount of variance explained by the personality traits.

It should be acknowledged and discussed why no placebo effect was induced with the paradigm in the control group. The authors may want to indicate in the limitation section that the majority of participants were female. Please also indicate the gender of the experimenters.

Please indicate the proportion of participants who were suspicious of the nature of the study depending on the group.

While the authors strictly follow the preregistered exclusion criteria which are good, it should at least be discussed that the final results may be prone to bias by excluding participants who did not develop the expectancy of analgesia, which is also a key outcome of the study. This would be particularly troublesome if the proportion of suspicious participants would differ between groups.

The authors may thus want to compliment an intention to treat like analysis showing the results of all participants.

*Reviewer #3:*

This study highlights an important potential confounding factor that might affect a wide range of clinical studies into personalized medicine. The authors show that the psychological effect of believing a procedure is personalized can modulate the perceived pain and unpleasantness of that procedure. The study is carefully designed but the effect sizes identified are quite small compared to the large inter-individual variability seen in all groups, raising questions about the generalizability of the results to other contexts. Furthermore, the authors test the psychological effect of personalization only on a subjective outcome, whereas many clinical studies in personalized medicine are focused on more objective measures such as disease survival that may be less affected by patients' belief in personalization.

The study involves two replicates, the first being a pilot/feasibility study involving 17 subjects that found a marginally significant decrease (38%) in perceived pain intensity in the group believing their test was personalized. The second was a pre-registered, double-blinded, placebo-controlled confirmatory study involving 85 patients. This second study measured a smaller effect (11%) than the first, but with stronger statistical support. The second study also identified a small (16%) effect on the perceived "unpleasantness" of pain. The second study identified a personality trait, the "need for uniqueness" as weakly moderating the effect of sham personalization on pain perception, as well as several other personality traits. In data aggregated across both experiments, pre-conditioning expectations correlated with pain perception.

The study appears well-designed. However, the consistent effects of sham personalization are quite small compared to the large differences in pain perception within both control and personalized groups, raising questions about how generalizable the study results will be. Though the statistical analysis shows statistically significant p-values, the larger confirmatory study yielded effect sizes smaller than the initial pilot study. One wonders whether in a larger third study, that might include for example several hundred individuals, the effect size might be smaller still. It is unclear from the manuscript as currently written how clinically significant an 11% decrease in perceived pain intensity might be. The manuscript would benefit from a better framing of the magnitude of the effects identified so that readers can more clearly understand the effects' practical significance.

The authors spent considerable effort designing a test that controls for a variety of potential confounding factors, developing an intricate sham personalization machine with bespoke equipment. However, by necessity the study involved consistent differences in communication between sham treatment and placebo groups, leaving open the possibility for uncontrolled confounding factors. For example, the additional interaction between subjects and study staff in the sham group might have altered the subjects' impression of their environment-a friendlier relationship with staff, an improved mood due to perceived kindness factors that might modulate pain tolerance independently from any specific belief in personalization. The possibility that such potential confounds might mediate in part the decrease in subjective pain intensity weakens the generalizability of the results to other contexts.

That said, the authors interpret their findings fairly narrowly, suggesting that clinical trials should be aware of the psychological effects of personalization. This seems like a solid recommendation irrespective of any technical limitations of the current study. However, it must be noted that the authors study the effect of belief in personalization only on subjective outcome-perceived pain intensity. One wonders about the relevance of these results for clinical work in personalized medicine focused primarily on objective outcomes, which may be less influenced by patients' beliefs during treatment.

I recommend that the authors frame better the magnitude of the effects identified so that readers can better understand their practical significance. I also recommend that the authors more explicitly the possibility that the experimental protocol for sham personalization might act not via a belief in personalization per se, but rather by modulating subjects' impression of their environment (more friendly) or by altering subjects' mood – two factors previously identified as modulating pain perception.

---

## [Author Response]

Essential revisions:Reviewers have outlined several recommendations for the authors. Below please find a summarized list with the essential revisions, but do refer to the reviewers' suggestions for details:1) Revise the terminology throughout the manuscript.

As described in more detail in the reviewer comments, we have now revised the terminology throughout the manuscript to clarify the distinction between active and sham treatments. We also, where possible, differentiated between them by using the term “intervention” when talking about active treatments.

2) Add missing details about the methodology and additional data regarding calibration procedures.

We elaborated on various aspects of methodology, such as the gender of the experimenters and the rates of suspicion in each group. We have also provided additional information regarding calibration procedures such as the average temperatures for each pain level in the sample of the confirmatory study.

3) Improve the graphical data presentation.

We have included additional graphs to show the individual changes in pain ratings and better demonstrate the variability in the raw scores. We have also included the graphs from the conditioning phase in the Appendix.

4) Add the statistical analysis on the existing data requested by the reviewers.

We clarify that the statistical analysis was already performed on raw scores, and therefore accounts for the random variation in pain ratings in the Off condition, suggested by Reviewer 1. We also discuss the importance of using per-protocol analysis instead of intention-to-treat analysis given our population of interest and the presence of deception. In brief, we suggest that our population of interest is only the participants who believe their treatment is personalised to them, and therefore those who are suspicious of our manipulation should be excluded following our pre-registered criteria.

5) Rework the discussion and reassess the extent of the claims taking into account the reviewers' suggestion, particularly with regards to the magnitude of the effects and its clinical significance, and the potential confounders that are not currently discussed (sample bias, intervening variables, etc.).

We have now reworked the discussion and expanded it on several points. First, we better contextualised our findings in terms of their clinical significance and relevance for clinical settings based on current guidelines. Second, we discussed in more detail several possible limitations such as the presence of sample bias and intervening variables. Finally, we expanded our methodology description and discussed the possibility of confounds suggested by Reviewer 3.

Reviewer #1:In this manuscript, Sandra et al. aimed at quantifying the role of the placebo effects of personalization in a randomized clinical trial.The main strengths of the manuscript are:– It presents data from an exploratory and a confirmatory study to test the same hypothesis.– The study presents data from several relevant variables that appear to have been carefully collected.The main weaknesses of the manuscript are:– The sample is not representative of the general population and the experimental settings are not a good match for clinical settings, which hinders the generalizability of the results.– The interpretation of the results does not consider potential implications related to individual vs group differences, or the experimental or clinical relevance of the effect sizes observed.I believe that the authors partially succeed in their aim, given that they are able to show a group effect of personalization in the quantification of the placebo effect. I believe that the discussion would benefit from contextualizing these results in the experimental settings, and reappraising them in relation to their actual clinical relevance.TerminologyThis might sound like a minor semantic detail, but the authors state that "precision treatments benefit from contextual factors", and that "treatment effects can be boosted by personalization", and this phrasing considers the placebo effect as part of the treatment. If I might suggest a different phrasing, I would say that the outcomes of an "intervention" can be constructed as the sum of the "real" treatment effect (if any) plus the placebo effect, and personalization in the context of this study only boosts the placebo effect. Here, the word "treatment" is used with a double meaning: as the action of attending to the patient's needs (what I suggest calling the intervention), and as the active component of the therapy that is supposed to produce physiological effects through mechanisms other than the placebo effect, that is, the treatment (which could be a pain-relief drug or a procedure such as TENS). Whereas this is not wrong per se, I would argue that the manuscript would benefit from clearly differentiating between these two terms, because this study does not have an active "treatment" condition, i.e., a group in which analgesia was provided by means known to work directly at the nociceptive system level. This would also be helpful to handle ethical issues that might arise if one considers placebo as part of the treatment: under these conclusions, pseudo-scientific therapies not backed by evidence (e.g., homeopathy) could be called "effective" just by "tailoring" or "personalizing" some aspect of the treatment to specific patient traits.

Thank you for this important suggestion. We have now clarified the distinction between the placebo effect and active ingredients throughout the manuscript, especially in the abstract and the introductory paragraphs. We have also used “intervention” when discussing precision medicine treatments to distinguish them from sham treatments:

Abstract

“Tailoring interventions to patient subgroups can improve intervention outcomes for various conditions. However, it is yet unclear how much of this improvement is due to the pharmacological personalisation itself versus the non-specific effects of the contextual factors involved in the tailoring process, such as therapeutic interaction. Here, we tested whether presenting a placebo analgesia machine as personalised would improve its effectiveness..… Conclusions. We present some of the first evidence that framing a sham treatment as personalised increases its effectiveness. Our findings could potentially improve the methodology of precision medicine research or inform practice.”

Introduction

“However, the greater effectiveness of tailored interventions may be due to more than just their pharmacological ingredients: *contextual factors*, such as the treatment setting and patient beliefs, may also directly contribute to better outcomes.”

In line with this, the use of the term "treatment effectiveness" might not be entirely adequate to refer to the placebo analgesia effects. A suggestion would be to qualify it as "perceived effectiveness", or to rephrase it as "Presenting a sham treatment as personalized increases the placebo effect in an RCT".

We have now revised the title to: “Presenting a Sham Treatment as Personalised Increases the Placebo Effect.”

Furthermore, there is no real "pain relief": as the figures show, there are changes in pain intensity ratings due to the summed effects of placebo and reduction in stimulation intensity.

We have used the term “pain relief” to refer to the reduced pain perception that follows from an intervention (real or placebo). If we understand correctly the reviewer’s comment, it seems that he would prefer to reserve the term “relief” to the “real” effects of an active treatment. We have now changed the term “pain relief” to “pain reduction” to refer to our findings.

To be clear, participants did experience pain reduction in the testing block of the procedure. During that part, we maintained the stimulation temperature at 50 level pain in the feasibility study and 40 level pain in the confirmatory study for all trials (when the machine was turned on and off) and for all participants. Therefore, any changes participants reported in their pain levels constituted pain reduction due to the placebo effect.

The true reduction in heat stimulation temperature happened only in the conditioning phase, which we did not include in the analyses.

Experimental designSettings and samplingI believe that the settings and sampling could have been improved. Even though the authors clearly state that volunteers were dropped if they asked about or mentioned the placebo effect, all participants are sampled from the McGill University community, and from the data, most are women who are also undergraduate students. Furthermore, it is fair to assume that the experiments were carried out in a laboratory at the Dept. of Psychology. In most circumstances (and for the feasibility study presented here) one would accept that an exploratory study might be carried out in these settings. However, the confirmatory study deals with a well-known psychological effect, is carried out in the Dept. of Psychology by psychologists, the volunteers are mostly sampled from psychology students, and the experiment has a large number of psychological questions related to the expectation of pain.

Thank you for this observation. We have now clarified that the testing settings were presented in a medical context. Indeed, the experimenters introduced themselves as neuroscience researchers; the study took place in a medical building that was located in a different building away from the psychology area.

Once at the lab, participants met two female experimenters at a medical building of a large Canadian university. The experimenters introduced themselves as neuroscience researchers and explained the study procedure.”

We also clarify that our sample for the confirmatory study was more diverse:

“We recruited 106 healthy participants aged 18 to 35 from the McGill University community; these were students and recent graduates from various disciplines.”

Furthermore, the "pain relief" device supposedly tailored to genetic information looks like technology from the 70s, when they could have just used the TENS stimulator that probably looks like cutting-edge, pain relief technology.Interestingly, the authors list many of these parts of the study as strengths, but I believe some of them are limitations.

The placebo machine used in the study was indeed “vintage” and could have looked more modern. However, we used it due to some unique features that we could not find on most TENS machines widely available today. The machine is large and clearly visible from most angles in the testing room, attracting attention in a way that a compact TENS may not. Additionally, it prominently displays over a dozen dials and switches, which we used to saliently “personalise” it to each participant. This likely helped emphasise the personalisation aspect of the procedure, despite the machine’s less-than-modern exterior. Overall, it is unclear whether another setting would have been more or less effective, but we do have a reliable inter-group difference in placebo effects from using it.

I actually find it odd that only 12% of participants having graduate training in biology, genetics, or psychology would not behave differently in this experiment (more on these below in the results subsection), and we actually have no data on how the general population would behave.

Indeed, the suspicion levels of the machine were low, and only 12% of the participants mentioned that the study focused on the placebo effect when probed for suspicion, and were excluded based on our pre-registered criteria. Although counterintuitive, this finding is consistent with other studies using complex deception that is now referenced in the manuscript:

“Only 12% of the participants questioned the veracity of the machine and none guessed the purpose of the study in the confirmatory study, despite many participants having graduate training in biology, genetics, or psychology. This is in line with previous studies on complex deception using intentionally elaborate placebos (Olson et al., 2016, 2020, 2023).”

The authors could have run the experiment on real medical settings with medical doctors (as opposed to psychology researchers) and a level of complexity during testing that mimics clinical routine, no more and no less (the abundance of psychological questions is particularly notorious).

We completely agree and are currently running another study in a hospital setting on chronic pain patients and using lidocaine IV infusion as a real treatment. Given the ethical concerns associated with studying placebo effect in patients, we first tested the presence of placebo effect of personalisation with experimental pain. This allowed us to control for the specific setting and pain levels to better isolate the effects of our manipulation. The positive findings from this study have now justified testing it in clinical settings despite the ethical concerns and will allow us to understand whether the more varied clinical pain is also subject to placebo effects of personalisation. We hope that this study will clarify the clinical significance of placebo effect of personalisation in treatment settings.

Sham conditionIf I followed the methodology correctly, the expected change in pain intensity in the conditioning stage of the feasibility study should be about 60 points for the feasibility study (80-20) and 40 points for the confirmatory study (60-20). The effect on pain intensity ratings is so large that participants should believe that this is nothing short of a miraculous machine (considering experimental settings, age, gender, and current health status). I suggest authors present data recorded on the observed effect in pain intensity and unpleasantness ratings during conditioning.

We have now included the data from the conditioning part of the study in the Appendix (Figure A3).

Following the experiment, participants proceed to the testing block, and the effect almost vanishes (an average of 5 points on pain intensity ratings, approximately 10% of the effect during the conditioning stage). I find it surprising that this fact apparently did not come up during the debriefing.

Indeed, several participants have mentioned that they thought the machine did not work for them at the end of the study; however, they did not doubt the veracity of the machine or the true nature of the study, and we thus kept them in the analyses.

Furthermore, debriefing is mentioned in the methods, but no results are reported in this regard.

We report the number of participants suspicious in the sample section of the study; however, we now also clarify the distribution of suspicion (and exclusion) by group:

“Of all participants, 1 did not complete the questionnaires which included the consent form, 6 did not fit eligibility criteria after consenting to participate, 1 experienced technical errors during the experiment, 1 refused to use the machine, and 12 mentioned or asked about the placebo effect (6 in each group).”

Interpretation of resultsI assume that the results shown in Figure 3 and 4 are the change in pain intensity and unpleasantness ratings during the testing block, for which the temperature at both the On and Off conditions is calibrated at a pain intensity of 50. I suggest reporting data from the calibration procedure in more detail, in order to compare it with the extensive knowledge available for thermal pain testing (e.g. Moloney et al., 2012).

We report more detail on calibration procedure, including the average temperatures for each pain level for the confirmatory study:

“On average, participants reported the pain threshold of 45.9 °C (*SD* = 1.7), as well as 46.9 °C (1.3) for pain level 20, 47.8 °C (1.0) for pain level 40, and 48.5 °C (1.2) for pain level 60.”

Furthermore, in the absence of a placebo effect (or other psychological confounders derived from the experimental settings), one would expect ratings during On and Off testing to be distributed around 50 points. In this regard, I suggest authors plot not just the differences, but the actual recorded ratings (with lines linking the On and Off data for each subject, since these are within-subject repeated measurements), and analyze the results using the Off condition as covariate (Vickers, 2001).

In our analysis model, we used raw pain ratings for intensity and unpleasantness, but for ease of presentation we plotted the changes per participant. We now include additional graph panels for each study that plot the individual raw pain ratings*.*

In line with this, differences between On and Off testing are to be within the measurement error for pain intensity ratings due to thermal stimulation, with an average difference between conditions of zero (no bias), as observed in the control condition. Since the authors report a non-zero average difference (5.8/100) they attribute this to the placebo effect.However, the settings for this experiment are quite particular: since it discusses "personalized" effects, it is important to pay attention to individual pain intensity and unpleasantness scores. From the figures, if one considers a binary scale, in which a difference of zero is no effect, any positive difference (regardless of its size) implies a placebo effect, and any negative difference (regardless of its size) implies a nocebo effect, then a significant number of participants is individually reporting nocebo effects in both conditions, i.e., they report more pain and unpleasantness during the On condition. This is surprisingly not mentioned anywhere in the text.

Thank you for your observation. We hesitate to discuss any non-zero difference between the “ON” and “OFF” conditions as indicative of a placebo or nocebo response. Indeed, we would expect that some variations would occur by chance – the difference will never be exactly zero – and we therefore cannot draw any strong conclusion at the individual level that someone is placebo or nocebo responder.

Nevertheless, we now mention the fact that not all participants experienced pain reductions in the “ON” condition.

“A number of participants in both groups also reported increases in pain ratings from using the machine.”

If, on the other hand, one is inclined to interpret these results as a group effect, the experimental/clinical relevance of the effect size must be considered, instead of just the statistical significance (Angst, 2017; de Vet et al., 2006). In this section of the Discussions, the authors should discuss the actual effect size (5.8/100) and its relation to the minimal clinically relevant effect. In my opinion, the detected group effect is too small to be considered clinically significant. The authors state "Our findings raise a question in the broad context of increasing interest in personalisation: just how big of a placebo effect is there from intervention tailoring? In this study, we show that the personalisation process was strong enough to influence the perception of thermal pain stimulations". While I agree that the personalization process somewhat influenced the perception of thermal pain thresholds at a group level, I also believe that this was only detectable with a relatively large sample, that the effect size is small (and in my opinion, not clinically significant), and that the implications of nocebo effects in individual pain ratings were not thoroughly analyzed and discussed.

Thank you for these interesting citations. Unfortunately, given our methodology, we are unable to use the clinical significance threshold suggested by Angst (2017). We do not have any qualitative measures of pain improvement in this study (e.g., the “slightly better” or the “about the same” qualifiers), but only numeric pain ratings. We agree that the effect sizes need to be discussed in context of clinical significance and expand on it below using medical recommendations.

Currently accepted IMMPACT recommendations for interpreting the importance of treatment effects make a clear distinction between what is meaningful at the individual versus group level (Dworkin et al., 2009). The typical guidelines when discussing clinically meaningful pain reduction usually refer to clinically meaningful difference at the *individual* level; these suggest that a reduction of 1/10 point or 10-20% represents a minimally important change. We observed an 11% reduction in pain intensity ratings and 16% reduction in pain unpleasantness, which places our results within this range at the *individual* level. Thus, even when using guidelines for assessing the importance of *individual* effects, it seems that our intervention is at least minimally clinically meaningful.

However, IMMPACT recommendations strongly suggest against using the same guidelines to judge the importance of group effects and instead recommends a case-by-case evaluation given the difference between clinical and experimental settings. Indeed, the therapeutic effect in clinical settings combines elements other than the active effects of the treatment, whereas the effects of randomised control trials only reveal the *incremental* effects of the active treatment compared to a control and are likely to be lower. Guidelines recommend a few strategies for assessing meaningfulness, such as comparing the effects to those of widely recognised treatments. Following these guidelines, the amplitude of our sham personalisation effects may be similar to the clinically significant effects of acetaminophen/paracetamol (Jürgens et al., 2014) or morphine (Koppert et al., 1999) on thermal pain perception. Our effect is particularly noteworthy given that the experimental and control condition were remarkably similar and only presented a subtle change in narrative of the placebo machine.

Thus, we believe that enhancing the personalisation aspects of treatment could have a clinically meaningful impact. However, we recognise that this remains to be tested in a clinical study.

We now include this additional paragraph in the discussion:

“Our effect was also small; the 11% reduction in pain intensity and 16% reduction in unpleasantness reached the lower threshold of minimal clinical significance of pain reduction (10 – 20%) suggested by guidelines (Dworkin et al., 2009). Nevertheless, testing placebo effects with experimental pain may have led to a conservative estimate of the placebo effect and may not map directly onto the clinical experience of chronic pain. Patients differ from healthy participants on many characteristics, including their motivation to get better(National Cancer Institute, 2021), the mechanisms through which they experience placebo effects (short- or long-term) (Vase et al., 2005), and the methods of assessing pain ratings (immediate versus retrospective). Our effect sizes were similar to that of paracetamol (Jürgens et al., 2014) and morphine (Koppert et al., 1999) on thermal pain, suggesting the possibility of clinical significance if tested in patients. Future studies could build on our proof-of-concept findings and explore whether these placebo effects apply to clinical populations who receive real personalised treatments focused on more objective measures. These additional investigations will help determine the clinical significance of placebo effect of personalisation for active treatments.

We have also changed the sentence mentioned above in the *Ideas and Speculation* section to reflect a narrower interpretation of the clinical significance of our data:

“Our results raise the possibility that placebo effects involved in personalisation may play a clinically relevant role in the broad context of the growing interest in precision medicine. In this study, we show that the personalisation process was strong enough to influence the perception of thermal pain.”

Reviewer #2:This pre-registered confirmatory study provides experimental evidence that the 'personalisation' of a treatment represents an important contextual factor that can enhance the placebo analgesic effect. This is shown in a well-controlled, double-blind study, where a sham analgesic treatment is either presented as "personalized" or not in 85 healthy volunteers. The authors show that volunteers in the 'personalized' group display stronger placebo analgesic responses as compared to the control group (although it should be noted that overall no placebo analgesic effect is induced in the control group, which is somewhat surprising given the conditioning procedure used). No differences in side effect profiles were observed, however, side effects were generally low in both groups, so the absence of differences between groups might be explained by a floor effect.Interestingly, these differences in placebo analgesia depending on 'personalisation' are not paralleled by significant differences in expectancy levels between groups neither before nor after conditioning. The authors also explore the role of different personality traits that may modulate the effect of 'personalisation' with interesting findings.Overall, the manuscript is very well written, and the rationale of the study is clear and highly relevant, and innovative in the context of the increasing attempts to individually tailor treatments, also referred to as precision medicine. The methods of the study are sound and the conclusions of the authors are adequately backed up by the data.These proof-of-concept findings obtained in healthy volunteers may have broad implications for the design and interpretation of clinical trials as well as systematic attempts to optimize treatment effects in clinical settings in the field of pain and beyond. Next studies have to test, whether the results translate into clinical contexts and beyond placebo analgesia.

Thank you for your comments.

The authors should indicate the amount of variance explained by the personality traits.

We now indicate the correlations of each personality trait with each group on the graphs in the main text and in the Appendix*.*

It should be acknowledged and discussed why no placebo effect was induced with the paradigm in the control group.

We now discuss this point in limitations in the context of potential mechanisms of action:

“The mechanism behind the placebo effect of personalisation may thus rely on an interaction with additional elements that need to be explored, for instance increases in mood from receiving a personalised treatment. It is also possible that the more complex mechanism is responsible for the general lack of placebo effect in the control group, but not the experimental group."

The authors may want to indicate in the limitation section that the majority of participants were female.

Indicated:

“Still, the main limitations of the study are its focus on healthy participants, the use of an inactive treatment, a sample with imbalanced genders, and the focus on a primarily subjective outcome of pain.”

Please also indicate the gender of the experimenters.

Indicated:

“Once at the lab, participants met two female experimenters introduced as neuroscience researchers at a medical building of a large Canadian university.”

Please indicate the proportion of participants who were suspicious of the nature of the study depending on the group.

Indicated:

“Of all participants, 1 did not complete the questionnaires which included the consent form, 6 did not fit eligibility criteria after consenting to participate, 1 experienced technical errors during the experiment, 1 refused to use the machine, and 12 mentioned or asked about the placebo effect (6 in each group).”

While the authors strictly follow the preregistered exclusion criteria which are good, it should at least be discussed that the final results may be prone to bias by excluding participants who did not develop the expectancy of analgesia, which is also a key outcome of the study. This would be particularly troublesome if the proportion of suspicious participants would differ between groups.

Thank you for the observation. Indeed, excluding participants who did not develop the expectancy of analgesia could increase bias and affect our results by artificially inflating the effect size. To avoid this issue, we strictly followed our pre-registered exclusion criteria: we only excluded participants who explicitly mentioned the placebo effect as the focus of the study with further elaboration. All other participants who merely thought the machine did not work for them or were disappointed by it were still included in the analyses reported here. We now clarify this in the main text:

“We were stringent with exclusion criteria to avoid positively biasing our effect: we only excluded participants who explicitly mentioned the placebo effect with additional explanations. For instance, one participant expressed general suspicion about stimulation timings and asked about placebo effects in the beginning of the session. The final sample included 85 participants (71 women) with a mean age of 21.4 (*SD* = 2.2).”

The authors may thus want to compliment an intention to treat like analysis showing the results of all participants.

We strictly followed our pre-registered exclusion criteria, thus reducing the possibility of bias and followed a per-protocol approach to analysis given the nature of our study. This is because intention-to-treat analysis is usually geared toward assessing the practical impact of the treatment in clinical settings and would be dealing with a different type of population. For instance, we are interested in the population that believes they are receiving a personalised treatment, as patients in clinical settings would. Therefore, participants who were suspicious of personalisation or the machine in the experimental study did not pass the essential manipulation check and cannot be comparable to the rest of the participants that represented our targeted population. This justified excluding them from the analyses. Were we to include all participants in our analyses (short of the technical malfunctions), all the *p* values would have increased to *p* >.05 for both measures. Nevertheless, this is not surprising, and there is little that we can conclude from this discrepancy in effects. Guessing the fact that the study focused on the placebo effect meant that these participants failed the crucial manipulation check of the experiment. They would therefore not be considered as the population of interest, which constitutes of people who trust a treatment they think is personalised to them.

We also suggest studying clinical populations in the discussion to better contextualise the relevance of the placebo effect for personalised medicine:

“Future studies could build on our proof-of-concept findings and explore whether these placebo effects apply to clinical populations who receive real personalised treatments focused on more objective measures.”

Reviewer #3:This study highlights an important potential confounding factor that might affect a wide range of clinical studies into personalized medicine. The authors show that the psychological effect of believing a procedure is personalized can modulate the perceived pain and unpleasantness of that procedure. The study is carefully designed but the effect sizes identified are quite small compared to the large inter-individual variability seen in all groups, raising questions about the generalizability of the results to other contexts. Furthermore, the authors test the psychological effect of personalization only on a subjective outcome, whereas many clinical studies in personalized medicine are focused on more objective measures such as disease survival that may be less affected by patients' belief in personalization.The study involves two replicates, the first being a pilot/feasibility study involving 17 subjects that found a marginally significant decrease (38%) in perceived pain intensity in the group believing their test was personalized. The second was a pre-registered, double-blinded, placebo-controlled confirmatory study involving 85 patients. This second study measured a smaller effect (11%) than the first, but with stronger statistical support. The second study also identified a small (16%) effect on the perceived "unpleasantness" of pain. The second study identified a personality trait, the "need for uniqueness" as weakly moderating the effect of sham personalization on pain perception, as well as several other personality traits. In data aggregated across both experiments, pre-conditioning expectations correlated with pain perception.The study appears well-designed. However, the consistent effects of sham personalization are quite small compared to the large differences in pain perception within both control and personalized groups, raising questions about how generalizable the study results will be. Though the statistical analysis shows statistically significant p-values, the larger confirmatory study yielded effect sizes smaller than the initial pilot study. One wonders whether in a larger third study, that might include for example several hundred individuals, the effect size might be smaller still. It is unclear from the manuscript as currently written how clinically significant an 11% decrease in perceived pain intensity might be. The manuscript would benefit from a better framing of the magnitude of the effects identified so that readers can more clearly understand the effects' practical significance.The authors spent considerable effort designing a test that controls for a variety of potential confounding factors, developing an intricate sham personalization machine with bespoke equipment. However, by necessity the study involved consistent differences in communication between sham treatment and placebo groups, leaving open the possibility for uncontrolled confounding factors. For example, the additional interaction between subjects and study staff in the sham group might have altered the subjects' impression of their environment-a friendlier relationship with staff, an improved mood due to perceived kindness factors that might modulate pain tolerance independently from any specific belief in personalization. The possibility that such potential confounds might mediate in part the decrease in subjective pain intensity weakens the generalizability of the results to other contexts.That said, the authors interpret their findings fairly narrowly, suggesting that clinical trials should be aware of the psychological effects of personalization. This seems like a solid recommendation irrespective of any technical limitations of the current study. However, it must be noted that the authors study the effect of belief in personalization only on subjective outcome-perceived pain intensity. One wonders about the relevance of these results for clinical work in personalized medicine focused primarily on objective outcomes, which may be less influenced by patients' beliefs during treatment.

We now put our findings in context and discuss their clinical significance:

“Our effect was also small; the 11% reduction in pain intensity and 16% reduction in unpleasantness reached the lower threshold of minimal clinical significance of pain reduction (10 – 20%) suggested by guidelines (Dworkin et al., 2009). Nevertheless, testing placebo effects with experimental pain may have led to a conservative estimate of the placebo effect and may not map directly onto the clinical experience of chronic pain. Patients differ from healthy participants on many characteristics, including their motivation to get better(National Cancer Institute, 2021), the mechanisms through which they experience placebo effects (short- or long-term) (Vase et al., 2005), and the methods of assessing pain ratings (immediate versus retrospective). Our effect sizes were similar to that of paracetamol (Jürgens et al., 2014) and morphine (Koppert et al., 1999) on thermal pain, suggesting the possibility of clinical significance if tested in patients. Future studies could build on our proof-of-concept findings and explore whether these placebo effects apply to clinical populations who receive real personalised treatments focused on more objective measures. These additional investigations will help determine the clinical significance of placebo effect of personalisation for active treatments.

We also have marked the focus on subjective outcomes in the limitations:

“Still, the main limitations of the study are its focus on healthy participants, the use of an inactive treatment, a sample including predominantly women, and the focus on a primarily subjective outcome of pain. […] Future studies could explore whether placebo effect of personalisation applies to clinical populations and real treatments focused on more objective measures; it could also determine the magnitude of effect in clinical settings.”

I recommend that the authors frame better the magnitude of the effects identified so that readers can better understand their practical significance.

We have improved the framing of the magnitude of the effects throughout the discussion, given the current evidence on clinical and practical significance (see previous comment).

We have also elaborated on the usefulness of our findings for clinical trials of personalised medicine:

“Our findings provide some of the first evidence for this novel placebo effect of personalisation and suggest its further study in clinical contexts, echoing experts in the field (Haga et al., 2009). It also supports the need for more consistent use of blinding, inactive control groups, and randomisation, especially for pivotal trials determining FDA approval of precision drugs (Pregelj et al., 2018). Indeed, only half of the FDA-approved precision treatments in recent years were based on double- or single-blinded pivotal trials, and only 38% of all pivotal trials used a placebo comparator (Pregelj et al., 2018). Although precision treatments are often developed for difficult-to-study diseases, their potential to elicit stronger placebo effects calls for more robust research designs.”

I also recommend that the authors more explicitly the possibility that the experimental protocol for sham personalization might act not via a belief in personalization per se, but rather by modulating subjects' impression of their environment (more friendly) or by altering subjects' mood – two factors previously identified as modulating pain perception.

Thank you for your recommendation. We would like to clarify that we paid careful attention when designing the study to avoid this confound and dedicated the same amount of time and attention to the participants in both groups. We now specify it further:

“To match the duration of participant interaction and explanations provided in the personalised group in an effort to reduce potential confounds, the experimenter instead described the different kinds of analgesics currently used in hospitals. The experimenter provided approximately 300 words of information to each group (280 in experimental and 298 in control). Finally, the experimenter introduced the machine with the same description and demonstration.”

However, we also mention the potential role that mood may play as an additional unmeasured mechanism or mediator of the study in limitations:

“The mechanism behind these placebo effects may thus rely on an interaction with additional elements that need to be explored, for instance increases in mood caused by receiving a personalised treatment.”